# EXPLOITING MINIMUM-VARIANCE POLICY EVALUATION FOR POLICY OPTIMIZATION

## ABSTRACT

Off-policy methods are the basis of a large number of effective Policy Optimization (PO) algorithms. In this setting, Importance Sampling (IS) is typically employed as a what-if analysis tool, with the goal of estimating the performance of a target policy, given samples collected with a different behavioral policy. However, in Monte Carlo simulation, IS represents a variance minimization approach. In this field, a suitable behavioral distribution is employed for sampling, allowing diminishing the variance of the estimator below the one achievable when sampling from the target distribution. In this paper, we analyze IS in these two guises, showing the connections between the two objectives. We illustrate that variance minimization can be used as a performance improvement tool, with the advantage, compared with direct off-policy learning, of implicitly enforcing a trust region. We make use of these theoretical findings to build a PO algorithm, Policy Optimization via Optimal Policy Evaluation (PO$^2$PE), that employs variance minimization as an inner loop. Finally, we present empirical evaluations on continuous RL benchmarks, with a particular focus on the robustness to small batch sizes.

## 1 INTRODUCTION

Policy Optimization methods (PO, Deisenroth et al., 2013) have been widely exploited in Reinforcement Learning (RL, Sutton & Barto, 2018) with successful results in addressing, to name a few, continuous-control (e.g., Peters & Schaal, 2008; Lillicrap et al., 2016), robot manipulation (e.g., Gu et al., 2017; Chatzilygeroudis et al., 2020), and locomotion (e.g., Kohl & Stone, 2004; Duan et al., 2016). Most of these algorithms employ the notion of *trust region* (Conn et al., 2000), introduced ante litteram in the RL literature by the *safe* RL approaches (Kakade & Langford, 2002; Pirotta et al., 2013), giving rise to a surge of effective algorithms, having TRPO (Schulman et al., 2015) as the progenitor. The core of any RL algorithm, being value-based or policy-based, lies in the ability to employ the samples collected with the current (or *behavioral*) policy to evaluate the performance of a candidate (or *target*) policy (Sutton & Barto, 2018). The skeleton rationale behind the usage of a trust region is to control the set of candidate policies whose performance can be accurately evaluated. Intuition suggests that if the candidate policy is "sufficiently close" to the current one, this *off-policy* evaluation problem (Precup et al., 2000) will provide a good estimate for the performance of the candidate policy. Formally, this idea has been studied in the field of Importance Sampling (IS, Owen, 2013) and the phenomenon is particularly evident looking at the IS estimator variance, which grows exponentially with the Rényi divergence (Rényi, 1961) between the behavioral and the target policy (Metelli et al., 2018; 2020). In this off-policy learning (Off-PL) setting, IS is employed as a *what-if* analysis tool (Owen, 2013) and its role is *passive*, as samples have been already collected with the current behavioral policy. In this sense, the trust region is an *a-posteriori* remedy for the limitations of off-policy evaluation, for controlling the uncertainty injected by the IS procedure.

However, IS originated in the Monte Carlo simulation community (Hesterberg, 1988; Hammersley, 2013) as an *active* tool for *variance minimization* (Off-VM). While in Off-PL, the behavioral policy is fixed and we look for the best target policy, whose performance we aim to estimate, here the roles are reversed. Indeed, in Off-VM, the target policy is fixed and we search for the behavioral policy (from which to collect samples) that yields an IS estimate with the minimum possible variance (Hammersley, 2013; Kahn & Marshall, 1953). It might seem surprising, at first, that sampling from a policy, other than the target one, can lead to an estimator with less variance (even zero in some cases) w.r.t. the on-policy estimate. In this role, IS has been previously employed in RL,

mainly to address rare events (Frank et al., 2008; Ciosek & Whiteson, 2017) which naturally lead to high-variance estimates, when tackled on-policy. The idea of explicitly using IS as a variance reduction technique, with the goal of finding an optimal behavioral policy, was proposed by (Hanna et al., 2017) for evaluation and subsequently combined with policy gradient learning (Hanna & Stone, 2018; Hanna et al., 2019). However, in these works, the variance minimization (Off-VM) process and the off-policy learning (Off-PL) problem are treated separately.

The goal of this paper is to investigate the relation between variance minimization (Off-VM) and off-policy learning (Off-PL). The core question we address can be summarized as: "*Can Off-VM be employed as a tool for Off-PL, overcoming the need for an explicit trust region?*" Intuitively, given a target policy, when the reward function is positive, one way to reduce the variance of the IS estimator is to assign larger probability to the trajectories that have a large impact on the mean, i.e., those with high returns. This provides a first hint about the connection between the minimum-variance sampling policy and the performance improvement, i.e., between Off-VM and Off-PL. Furthermore, it suggests that we could repeatedly apply the process of identifying the minimum-variance policy as a tool for policy improvement. The interesting aspect of such an approach is that, by minimizing the variance, it *implicitly* controls the divergence between two consecutive policies. In other words, it allows enforcing a trust region, without the need for divergence constraints or penalizations.

**Outline of the Contributions** In this paper, we provide theoretical, algorithmic, and experimental contributions. After having introduced the background (Section 2), we present the problem of finding the minimum-variance behavioral distribution (Section 3). Then, we study the properties of the Off-VM problem in two settings: unconstrained (Section 4) and constrained (Section 5). First, we assume that there are no restrictions for choosing the behavioral distribution. We show that the minimum-variance behavioral distribution, besides leading to the zero-variance estimator (Kahn & Marshall, 1953), is guaranteed to yield a performance improvement, requiring the non-negativity of the reward only. Furthermore, we prove that this approach allows controlling the divergence between two consecutive distributions, thus enforcing an implicit trust region. Although this provides a valuable starting point, the minimum-variance distribution might be unrealizable given the environment transition model, i.e., there might be no policy inducing it. For this reason, we move to the scenario in which the distributions are constrained in a suitable space. In this setting, the zero-variance estimator could not be achievable. Nevertheless, we prove that such a procedure can lead to a performance improvement and preserves the trust region enforcement. Based on these theoretical results, we propose *Policy Optimization via Optimal Policy Evaluation* (PO$^2$PE), a novel PO algorithm, that we particularize for parametric policy spaces (Section 6). Finally, we provide numerical simulations on continuous-control benchmarks, in comparison with POIS (Metelli et al., 2018) and TRPO (Schulman et al., 2015), with a particular focus on the robustness of PO$^2$PE to small batch sizes (Section 7). The proof of the results presented in the main paper are reported in Appendix A.

## 2 PRELIMINARIES

In this section, we provide the necessary background that will be employed in the paper.

**Mathematical Notation** Let $\mathcal{X}$ be a set, and let $\mathfrak{F}_\mathcal{X}$ be a $\sigma$-algebra over $\mathcal{X}$. We denote with $\mathscr{P}(\mathcal{X})$ the space of probability measures over $(\mathcal{X}, \mathfrak{F}_\mathcal{X})$. Let $P \in \mathscr{P}(\mathcal{X})$, whenever needed, we assume that $P$ admits a density function $p$. For a subset $\mathcal{Y} \subseteq \mathbb{R}$, we denote with $\mathscr{B}(\mathcal{X}, \mathcal{Y})$ the space of measurable functions $f : \mathcal{X} \to \mathcal{Y}$. Let $P, Q \in \mathscr{P}(\mathcal{X})$ be two probability measures such that $P \ll Q$, i.e., $P$ is absolutely continuous w.r.t. $Q$, for every $\alpha \in [0, \infty]$, we define the $\alpha$-*Rényi divergence* as (Rényi, 1961): $D_\alpha(P\|Q) = \frac{1}{\alpha-1} \log \int_\mathcal{X} p(x)^\alpha q(x)^{1-\alpha} \mathrm{d}x$. In the limit of $\alpha \to 1$, the Rényi divergence reduces to the KL-divergence $D_{\mathrm{KL}}(P\|Q)$, while for $\alpha \to \infty$, it corresponds to $\mathrm{ess\,sup}_{x \sim Q}\{p(x)/q(x)\}$.

**Importance Sampling** Let $P, Q \in \mathscr{P}(\mathcal{X})$ with $P \ll Q$ and let $f \in \mathscr{B}(\mathcal{X}, \mathbb{R})$. Importance Sampling (IS, Owen, 2013) allows estimating the expectation of $f$ under a *target* distribution $P$, i.e., $\mathbb{E}_{x \sim P}[f(x)]$ having samples $\{x_i\}_{i \in [n]}$ collected with a *behavioral* distribution $Q$: $\widehat{\mu}_{P/Q} = \frac{1}{n}\sum_{i \in [n]} \frac{p(x_i)}{q(x_i)} f(x_i)$. The IS estimator is unbiased (Owen, 2013), i.e., $\mathbb{E}_{x_i \sim Q}[\widehat{\mu}_{P/Q}] = \mathbb{E}_{x \sim P}[f(x)]$, but it might suffer from large variance, due to the heavy-tailed behavior (Metelli et al., 2018). The properties of $\widehat{\mu}_{P/Q}$ and its transformations have been extensively studied in the literature (e.g., Ionides, 2008; Thomas et al., 2015; Papini et al., 2019; Kuzborskij et al., 2021; Metelli et al., 2020).

**Policy Optimization** A Markov Decision Process (MDP, Puterman, 1994) is a 6-tuple $\mathcal{M} = (\mathcal{S}, \mathcal{A}, P, R, \gamma, D_0)$, where $\mathcal{S}$ is the state space, $\mathcal{A}$ is the action space, $P : \mathcal{S} \times \mathcal{A} \rightarrow \mathscr{P}(\mathcal{S})$ is the transition model, $R : \mathcal{S} \times \mathcal{A} \rightarrow [0, R_{\max}]$ is the reward function, $\gamma \in [0, 1]$ is the discount factor, and $D_0 \in \mathscr{P}(\mathcal{S})$ is the initial state distribution. The agent's behavior is modeled by a *parametric* policy $\pi_{\boldsymbol{\theta}} : \mathcal{S} \rightarrow \mathscr{P}(\mathcal{A})$ belonging to a parametric policy space $\Pi_\Theta = \{\pi_{\boldsymbol{\theta}} : \boldsymbol{\theta} \in \Theta \subseteq \mathbb{R}^d\}$. The interaction between an agent and the MDP generates a *trajectory* $\tau = (s_0, a_0, s_1, a_1, \ldots, s_{H-1}, a_{H-1}, s_H)$ where $H \in \mathbb{N}$ is the trajectory length and $s_0 \sim D_0$, $a_t \sim \pi_{\boldsymbol{\theta}}(\cdot|s_t)$, $s_{t+1} \sim P(\cdot|s_t, a_t)$ for all $t \in \{0, \ldots, H-1\}$. Given a trajectory $\tau$, the *return* is the discounted sum of the rewards $\mathcal{R}(\tau) = \sum_{t=0}^{H-1} \gamma^t R(s_t, a_t)$. For a policy $\pi_{\boldsymbol{\theta}} \in \Pi_\Theta$, we denote with $p(\cdot|\boldsymbol{\theta})$ the induced trajectory distribution: $p(\tau|\boldsymbol{\theta}) = D_0(s_0) \prod_{t=0}^{H-1} \pi_{\boldsymbol{\theta}}(a_t|s_t) P(s_{t+1}|s_t, a_t)$. In the *action-based* (AB) setting, an agent aims at finding a parametrization fulfilling: $\boldsymbol{\theta}^* \in \arg\max_{\boldsymbol{\theta} \in \Theta} \{J(\boldsymbol{\theta})\}$, where $J(\boldsymbol{\theta}) = \mathbb{E}_{\tau \sim p(\cdot|\boldsymbol{\theta})}[\mathcal{R}(\tau)]$ is the *expected return*. $\pi_{\boldsymbol{\theta}}$ must be stochastic to ensure exploration. Instead, in the *parameter-based* (PB) setting, we consider a *hyperpolicy* $\nu_{\boldsymbol{\rho}}$, belonging to a parametric hyperpolicy space $\mathcal{N}_\mathcal{P} = \{\nu_{\boldsymbol{\rho}} : \boldsymbol{\rho} \in \mathcal{P} \subseteq \mathbb{R}^l\}$, from which we sample the parameters $\boldsymbol{\theta}$ of the policy. In this case, the policy $\pi_{\boldsymbol{\theta}}$ can be deterministic since exploration is managed at the hyperpolicy level and the agent goal becomes to learn a hyperpolicy parametrization maximizing the expected return: $\boldsymbol{\rho}^* \in \arg\max_{\boldsymbol{\rho} \in \mathcal{P}} \{J(\boldsymbol{\rho})\}$, where $J(\boldsymbol{\rho}) = \mathbb{E}_{\boldsymbol{\theta} \sim \nu_{\boldsymbol{\rho}}}[J(\boldsymbol{\theta})]$. In the paper, we keep the presentation as general as possible, introducing the results for arbitrary distributions. Then, we will particularize for the parametric PO setting.

## 3 MINIMUM–VARIANCE BEHAVIORAL DISTRIBUTION

In this section, we revise Off-VM, i.e., the problem of finding a behavioral distribution $Q \in \mathscr{P}(\mathcal{X})$ that induces an IS estimate $\hat{\mu}_{P/Q}$ with minimum variance, knowing the (fixed) target distribution $P \in \mathscr{P}(\mathcal{X})$ and function $f \in \mathscr{B}(\mathcal{X}, [0, \infty))$.[1] Furthermore, we do not enforce any restriction on the possible forms of the behavioral distribution $Q \in \mathscr{P}(\mathcal{X})$. The problem and the corresponding well-known *minimum-variance behavioral distribution* $Q^*$ are stated in the following (Kahn, 1950):

$$\min_{Q \in \mathscr{P}(\mathcal{X})} \left\{ \mathrm{Var}_{x \sim Q} \left[ \frac{p(x)}{q(x)} f(x) \right] \right\} \quad \implies \quad q^*(x) = \frac{p(x)f(x)}{\mathbb{E}_{x \sim P}[f(x)]}, \quad \forall x \in \mathcal{X}. \tag{1}$$

We observe that the IS estimator $\hat{\mu}_{P/Q^*}$ is non-stochastic, equal to the quantity we aim to estimate, i.e., $\hat{\mu}_{P/Q^*} = \mathbb{E}_{x \sim P}[f(x)]$. This suggests that the construction of $Q^*$ is infeasible as it requires knowledge of $\mathbb{E}_{x \sim P}[f(x)]$. Since $Q^*$ generates a non-stochastic estimator, it not only leads to zero-variance but, clearly, simultaneously minimizes the absolute central moments of any order. A second, and most remarkable property, is that $Q^*$ is a *performance improvement* w.r.t. $P$, i.e., the expectation of $f$ under $Q^*$ is larger than the expectation of $f$ under the target $P$ (Owen, 2013):

$$\mathbb{E}_{x \sim Q^*}[f(x)] - \mathbb{E}_{x \sim P}[f(x)] = \frac{\mathrm{Var}_{x \sim P}[f(x)]}{\mathbb{E}_{x \sim P}[f(x)]} \geqslant 0. \tag{2}$$

It is worth noting that the magnitude of the improvement is directly related to the reduction in variance $\mathrm{Var}_{x \sim P}[f(x)]$. Equation (2) suggests an appealing connection between the problem of finding the minimum-variance behavioral distribution (Off-VM) and the problem of finding a target distribution that maximizes the expectation $\mathbb{E}_{x \sim P}[f(x)]$ (Off-PL). In other words, we could employ Off-VM as a performance improvement tool, by repeatedly solving the problem in Equation (1).

In the following two sections, we will delve into the properties of the repeated construction of the minimum-variance distribution as a performance improvement tool under two assumptions: (i) there are no restrictions in the choice of the behavioral distribution $Q \in \mathscr{P}(\mathcal{X})$ (Section 4); (ii) the behavioral distribution must be chosen within a subset $Q \in \mathcal{Q} \subseteq \mathscr{P}(\mathcal{X})$ (Section 5). In both cases, we will address the following three questions: **(Q1)** Does this procedure always generate a distribution that is a *performance improvement*? **(Q2)** Does this procedure *converge* to a (global or local) maximum of $f$? **(Q3)** Can we quantify the divergence between two consecutive distributions, i.e., does this procedure enforce a *trust region*?

Before proceeding, let us map this general setting to PO. In the action-based (AB) setting, $x$ is the trajectory $\tau$, $P$ and $Q$ are trajectory distributions $p(\tau|\boldsymbol{\theta})$ induced by policies $\pi_{\boldsymbol{\theta}}$. Instead, in the parameter-based (PB) setting, $x$ is the pair $(\boldsymbol{\theta}, \tau)$, $P$ and $Q$ are joint distributions $\nu_{\boldsymbol{\rho}}(\boldsymbol{\theta})p(\tau|\boldsymbol{\theta})$ induced by hyperpolicies $\nu_{\boldsymbol{\rho}}$. In both cases, function $f$ is the trajectory return $\mathcal{R}(\tau)$.

---

[1]We restrict our attention to non-negative functions. From the RL perspective, this choice is w.l.o.g. since we can always define an equivalent non-negative reward function, by means of a translation of the original one.

## 4 Unconstrained Probability Distribution Space

In Section 3, we have seen that $Q^*$ is a performance improvement w.r.t. $P$. We now generalize this construction, by composing function $f$ with a non-negative monotonic strictly-increasing function $h:[0,\infty)\to[0,\infty)$. The rationale behind this choice is that if $h$ is strictly-increasing, then $h\circ f$ has the same maxima as $f$.[2] We start defining the operator $\mathcal{I}_{h\circ f}:\mathscr{P}(\mathcal{X})\to\mathscr{P}(\mathcal{X})$:

$$\left(\mathcal{I}_{h\circ f}[P]\right)(x)=\frac{p(x)h(f(x))}{\mathbb{E}_{x\sim P}[h(f(x))]},\qquad \forall x\in\mathcal{X}. \tag{3}$$

Thus, $\mathcal{I}_{h\circ f}$ takes as input a target distribution $P\in\mathscr{P}(\mathcal{X})$, a function $h\circ f\in\mathscr{B}(\mathcal{X},[0,\infty))$, and outputs the minimum-variance behavioral distribution for the IS estimation of $\mathbb{E}_{x\sim P}[h(f(x))]$, i.e., $Q^*=\mathcal{I}_{h\circ f}[P]$. Intuitively, looking at Equation (3), by iterating the application of $\mathcal{I}_{h\circ f}$, we will obtain distributions tending to assign larger probability mass to points $x\in\mathcal{X}$ with high values of $f(x)$. Concerning **(Q1)**, the following result, due to Ghosh et al. (2020), generalizes Equation (2) showing that whenever $h$ is increasing, we can prove that $\mathcal{I}_{h\circ f}[P]$ is a performance improvement w.r.t. $P$.

**Proposition 4.1** (Proposition 9 of Ghosh et al. (2020)). *Let $P\in\mathscr{P}(\mathcal{X})$, $f\in\mathscr{B}(\mathcal{X},[0,\infty))$, and $h:[0,\infty)\to[0,\infty)$ monotonic increasing. Then, $\mathcal{I}_{h\circ f}[P]$ is a performance improvement w.r.t. $P$:*

$$\mathbb{E}_{x\sim\mathcal{I}_{h\circ f}[P]}[f(x)]-\mathbb{E}_{x\sim P}[f(x)]=\frac{\mathrm{Cov}_{x\sim P}[h(f(x)),f(x)]}{\mathbb{E}_{x\sim P}[h(f(x))]}\geqslant 0.$$

Note that, since $h$ is a monotonic increasing function, we have that $\mathrm{Cov}_{x\sim P}[h(f(x)),f(x)]\geqslant 0$ (Cuadras, 2002). The following sections tackle questions **(Q2)** and **(Q3)**.

### 4.1 Convergence Properties

We now address question **(Q2)**, analyzing the effect of repeatedly applying operator $\mathcal{I}_{h\circ f}$. More formally, let us consider an initial distribution $P\in\mathscr{P}(\mathcal{X})$, and suppose to iterate the application of the operator $\mathcal{I}_{h\circ f}$, generating the sequence of distributions $(Q_k)_{k\in\mathbb{N}}$, where $Q_0=P$ and for every $k\in\mathbb{N}_{\geqslant 0}$ we have $Q_k=\mathcal{I}_{h\circ f}[Q_{k-1}]=\left(\mathcal{I}_{h\circ f}\right)^k[P]$. The following result shows that, under certain conditions, the operator $\mathcal{I}_{h\circ f}$ admits fixed points and the sequence $(Q_k)_{k\in\mathbb{N}}$ converges to a distribution $Q_\infty$ that assigns probability only to the global maxima of $f$, restricted to the support of $P$, i.e., $\mathrm{supp}(P)$.

**Theorem 4.2.** *Let $P\in\mathscr{P}(\mathcal{X})$, $f\in\mathscr{B}(\mathcal{X},[0,\infty))$, and $h:[0,\infty)\to[0,\infty)$ monotonic strictly-increasing. Then, the following statements hold:*

*(i)* *$P$ is a fixed point of $\mathcal{I}_{h\circ f}$, i.e., $\mathcal{I}_{h\circ f}[P]=P$ a.s., if and only if $\mathrm{Var}_{x\sim P}[f(x)]=0$;*

*(ii)* *let $\mathcal{X}^*=\arg\max_{x\in\mathrm{supp}(P)}\{f(x)\}$ be the set of maxima of $f$ restricted to the support of $P$. If $\mathcal{X}^*$ is non-empty and measurable then, the repeated application of $\mathcal{I}_{h\circ f}$ converges to a distribution $Q_\infty=\lim_{k\to\infty}\left(\mathcal{I}_{h\circ f}\right)^k[P]$ with support $\mathcal{X}^*$. In particular $\mathbb{E}_{x\sim Q_\infty}[f(x)]=\max_{x\in\mathrm{supp}(P)}\{f(x)\}$.*

Some remarks are in order. First, both properties are independent of the function $h$ as long as it is non-negative and monotonic strictly-increasing. This is expected since, $h\circ f$ admits the same set of global optima of $f$. Second, as a corollary to point (i), any deterministic $P$ is a fixed point of $\mathcal{I}_{h\circ f}$. Finally, from point (ii), we deduce that if we select $P$ that assigns non-zero probability to all points in $\mathcal{X}$, i.e., $\mathrm{supp}(P)=\mathcal{X}$, the iterated application of $\mathcal{I}_{h\circ f}$ converges to the distribution $Q_\infty$ such that $\mathbb{E}_{x\sim Q_\infty}[f(x)]=\max_{x\in\mathcal{X}}\{f(x)\}$, i.e., we are performing a global optimization of $f$.

### 4.2 Implicit Trust Region

The reader might wonder what are the advantages of casting the optimization of function $f$ as such an iterative procedure. The reason lies in question **(Q3)**. We now prove that we are able to naturally control the divergence between two consecutive distributions $Q_k$ and $Q_{k+1}=\mathcal{I}_{h\circ f}[Q_k]$ with $k\in\mathbb{N}$, with the effect of enforcing an *implicit* trust region. The following result shows how it is possible to obtain a bound on the $\alpha$-Rényi divergence between two consecutive distributions.

---

[2] As we shall see in the following sections, the different choices of $h$ will be useful to control the trust region of the optimization process.

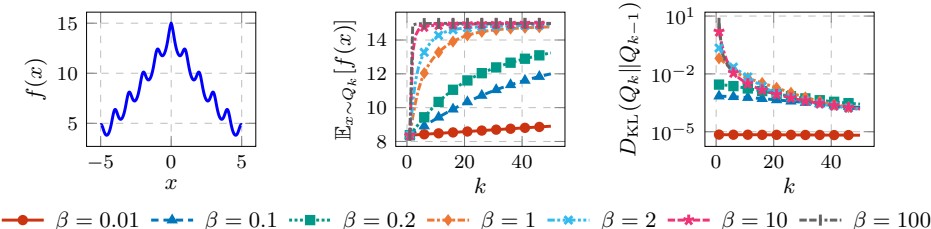

Figure 1: The Ackley function (left), the expectation of the distribution $Q_k = (\mathcal{I}_{h \circ f})^k[P]$ (center), and the KL-divergence (right) between two consecutive distributions $Q_{k-1}$ and $Q_k$, with $h = (\cdot)^\beta$.

**Theorem 4.3.** *Let $P \in \mathscr{P}(\mathcal{X})$, $f \in \mathscr{B}(\mathcal{X}, [0, \infty))$, and $h : [0, \infty) \to [0, \infty)$ monotonic strictly-increasing. Then, for every $\alpha \in [0, \infty]$, it holds that:*

$$D_\alpha\left(\mathcal{I}_{h \circ f}[P] \| P\right) = \frac{1}{\alpha - 1} \log \frac{\mathbb{E}_{x \sim P}[h(f(x))^\alpha]}{\mathbb{E}_{x \sim P}[h(f(x))]^\alpha}.$$

*In particular, for $\alpha = 1$ it holds that:*

$$D_{KL}(\mathcal{I}_{h \circ f}[P] \| P) = \frac{\mathrm{Cov}_{x \sim P}[h(f(x)), \log h(f(x))]}{\mathbb{E}_{x \sim P}[h(f(x))]}.$$

For $\alpha = 2$, we obtain $D_2(I_{h \circ f}[P] \| P) = \log \frac{\mathbb{E}_{x \sim P}[h(f(x))^2]}{\mathbb{E}_{x \sim P}[h(f(x))]^2} \leqslant \frac{\mathrm{Var}_{x \sim P}[h(f(x))]}{\mathbb{E}_{x \sim P}[h(f(x))]^2}$. Thus, the divergence is large when the variance of $h(f(x))$ is. The result is particularly remarkable as we are able to control the Rényi divergences of *any* order $\alpha \in [0, \infty]$. This is a relevant achievement since the trust regions commonly used, like KL-divergence (Schulman et al., 2015), are unable to control higher-order divergences that can still be infinite. We can also appreciate the role of the increasing function $h$ that works as a regularizer with the effect of controlling the size of the trust region. The following example shows that the faster $h$ increases, the larger the induced trust region becomes.

**Example 4.1.** *We consider (a slight variation of) the one-dimensional Ackley function (Ackley, 2012): $f(x) = -5 + 20 \exp(-0.1414|x|) + \exp(0.5(\cos(2\pi x) + 1)) + e$, shown in Figure 1 (left) and the class of increasing functions $(h \circ f)(x) = f(x)^\beta$ where $\beta \geqslant 0$. We consider an initial uniform distribution $P = \mathrm{Uni}([-5, 5])$. In Figure 1, we plot the expectation of distribution $Q_k = (\mathcal{I}_{h \circ f})^k[P]$ (center) and the KL-divergence between two consecutive distributions (right), as a function of the number of applications $k$, for the different $\beta$ values. We observe that convergence to the global optimum ($x^* = 0$ and $f(x^*) = 15$) is faster for higher powers that also lead to larger trust regions.*

## 5 CONSTRAINED PROBABILITY DISTRIBUTION SPACE

The approach we have presented in Section 4 can be applied when there are *no* restrictions on the class of distributions that can be played, i.e., we can select $Q$ in the whole space $\mathscr{P}(\mathcal{X})$. However, in the action-based PO, we can intervene on the policy $\pi_\theta$ factors only of the distribution $p(\tau | \theta) = D_0(s_0) \prod_{t=0}^{H-1} \pi_\theta(a_t | s_t) P(s_{t+1} | s_t, a_t)$, leading to a constrained setting. Similarly, in the parameter-based PO, we can act on the hyperpolicy $\nu_\rho$ while keeping the trajectory distribution $p(\tau | \theta)$ fixed.

More in general, when considering a class of distributions $\mathcal{Q} \subseteq \mathscr{P}(\mathcal{X})$, even if $P \in \mathcal{Q}$, the distribution $\mathcal{I}_{h \circ f}[P]$ might not belong to $\mathcal{Q}$. Furthermore, while $\mathcal{I}_{h \circ f}[P]$ minimizes *all* absolute central $\alpha$-moments of the IS estimator, as it leads to a non-stochastic estimator (Section 3), there may exist different distributions in $\mathcal{Q}$ minimizing the different absolute central $\alpha$-moments:

$$\min_{Q \in \mathcal{Q}} \left\{ \mathbb{E}_{x \sim Q} \left[ \left| \frac{p(x)}{q(x)} h(f(x)) - \mathbb{E}_{x \sim P}[h(f(x))] \right|^\alpha \right] \right\}. \tag{4}$$

Apart from $\alpha = 2$, where the problem in Equation (4) reduces to Equation (1), for general value of $\alpha \in [0, \infty]$, the optimization is not straightforward (e.g., Equation (4) is not differentiable for $\alpha \in (0, 2)$). The following result shows that performing a *moment projection* through the $\alpha$-Rényi divergence is a reasonable surrogate for minimizing the absolute central $\alpha$-moments of Equation (4).

**Proposition 5.1.** *Let $P \in \mathscr{P}(\mathcal{X})$, $f \in \mathscr{B}(\mathcal{X}, [0, \infty))$, and $h : [0, \infty) \to [0, \infty)$ monotonic strictly-increasing. Then, for any $\alpha \in (1, \infty)$, it holds that:*

$$\underbrace{\mathbb{E}_{x\sim Q}\left[\left|\frac{p(x)}{q(x)}h(f(x))-\mathbb{E}_{x\sim P}[h(f(x))]\right|^{\alpha}\right]}_{\textit{absolute central }\alpha\textit{-moment}}\leqslant\underbrace{\mathbb{E}_{x\sim Q}\left[\left(\frac{p(x)}{q(x)}h(f(x))\right)^{\alpha}\right]}_{\textit{(non-central) }\alpha\textit{-moment}}=e^{(\alpha-1)D_{\alpha}(\mathcal{I}_{h\circ f}[P]\|Q)}\mathbb{E}_{x\sim P}[h(f(x))]^{\alpha}.$$

Thus, having considered the subset of distributions $\mathcal{Q}\subseteq\mathscr{P}(\mathcal{X})$, whenever $\mathcal{I}_{h\circ f}[P]\notin\mathcal{Q}$, we replace it with the corresponding moment projection performed through the $\alpha$-Rényi divergence:

$$Q^{\dagger}\in\underset{Q\in\mathcal{Q}}{\arg\min}\left\{D_{\alpha}(\mathcal{I}_{h\circ f}[P]\|Q)\right\}. \tag{5}$$

In the following, we shall address the questions **(Q1)**, **(Q2)**, and **(Q3)** for the constrained setting.

## 5.1 Performance Improvement

In Proposition 4.1, we have seen that, whenever $h$ is strictly-increasing, $\mathcal{I}_{h\circ f}[P]$ is a performance improvement w.r.t. $P$, evaluated under function $f$ (and also under the composition between $f$ and *any* strictly-increasing function). In this section, we address question **(Q1)**, showing that, when considering a subset of distributions $\mathcal{Q}\subseteq\mathscr{P}(\mathcal{X})$, the performance improvement cannot be in general guaranteed for $f$, but just for a *specific* monotonic transformation of $f$, depending on $h$ and $\alpha$.

**Theorem 5.2.** *Let $P\in\mathscr{P}(\mathcal{X})$, $f\in\mathscr{B}(\mathcal{X},[0,\infty))$, and $h:[0,\infty)\to[0,\infty)$ monotonic strictly-increasing. Let $\mathcal{Q}\subseteq\mathscr{P}(\mathcal{X})$, $Q\in\mathcal{Q}$, and $\alpha\in[0,\infty]$, then, it holds that:*

$$\mathbb{E}_{x\sim Q}\left[h(f(x))^{\alpha}\right]-\mathbb{E}_{x\sim P}[h(f(x))^{\alpha}]\geqslant\frac{\mathbb{E}_{x\sim P}[h(f(x))]^{\alpha}}{\alpha-1}\left(e^{(\alpha-1)D_{\alpha}(\mathcal{I}_{h\circ f}[P]\|P)}-e^{(\alpha-1)D_{\alpha}(\mathcal{I}_{h\circ f}[P]\|Q)}\right).$$

*In particular, for $\alpha=1$, it holds that (Ghosh et al., 2020, Proposition 6):*

$$\mathbb{E}_{x\sim Q}\left[h(f(x))\right]-\mathbb{E}_{x\sim P}[h(f(x))]\geqslant\mathbb{E}_{x\sim P}[h(f(x))]\left(D_{KL}(\mathcal{I}_{h\circ f}[P]\|P)-D_{KL}(\mathcal{I}_{h\circ f}[P]\|Q)\right).$$

The theorem shows that by minimizing the $\alpha$-moment of the transformed function $h\circ f$, we are able to guarantee a performance improvement on the function $(\cdot)^{\alpha}\circ h\circ f$. The result holds provided that $D_{\alpha}(\mathcal{I}_{h\circ f}[P]\|Q)\leqslant D_{\alpha}(\mathcal{I}_{h\circ f}[P]\|P)$, which is always guaranteed when $P\in\mathcal{Q}$ and $Q=Q^{\dagger}$, being $Q^{\dagger}$ defined in Equation (5) as the minimizer of the second divergence term. In particular, if we select $h=(\cdot)^{1/\alpha}$, the guarantee holds for the function $f$ directly. For all other choices, the performance improvement can be guaranteed for a monotonic transformation of $f$ only.[3]

## 5.2 Convergence Properties

We now turn to **(Q2)**. By using Equation (5) as an iterate $Q_{k+1}\in\arg\min_{Q\in\mathcal{Q}}\left\{D_{\alpha}(\mathcal{I}_{h\circ f}[Q_k]\|Q)\right\}$ to generate a sequence of distributions $(Q_k)_{k\in\mathbb{N}}$, we are *not* guaranteed to converge to any fixed-point distribution $Q_{\infty}$, differently form the unconstrained setting (Theorem 4.2). This is because the minimization might yield multiple solutions. Nevertheless, we are able to provide guarantees on the final divergence value and on the performance of the distributions $Q_k$.

**Theorem 5.3.** *Let $P\in\mathscr{P}(\mathcal{X})$, $f\in\mathscr{B}(\mathcal{X},[0,\infty))$, and $h:[0,\infty)\to[0,\infty)$ monotonic strictly-increasing. Let $\mathcal{Q}\subseteq\mathscr{P}(\mathcal{X})$ and suppose that $h\circ f$ is bounded from above, then, the iterate $Q_{k+1}\in\arg\min_{Q\in\mathcal{Q}}\left\{D_{\alpha}(\mathcal{I}_{h\circ f}[Q_k]\|Q)\right\}$ (where possible ties are broken arbitrarily) satisfies:*

*(i) the sequence of divergences $D_{\alpha}(\mathcal{I}_{h\circ f}[Q_k]\|Q_k)$ is convergent;*
*(ii) the sequence of expectations $\mathbb{E}_{x\sim Q_k}\left[h(f(x))^{\alpha}\right]$ is non-decreasing in $k\in\mathbb{N}$ and converges to a stationary point of $\mathbb{E}_{x\sim Q}\left[h(f(x))^{\alpha}\right]$ w.r.t. $Q\in\mathcal{Q}$.*

The convergence of the sequences $D_{\alpha}(\mathcal{I}_{h\circ f}[Q_k]\|Q_k)$ and $\mathbb{E}_{x\sim Q_k}\left[h(f(x))^{\alpha}\right]$ is derived by the performance improvement of Theorem 5.2. Importantly, Theorem 5.3 shows the convergence to a *stationary point* of $\mathbb{E}_{x\sim Q}\left[h(f(x))^{\alpha}\right]$. If $\mathcal{Q}$ is a parametric space $\mathcal{Q}_{\Xi}=\{Q_{\xi}\in\mathscr{P}(\mathcal{X}):\xi\in\Xi\subseteq\mathbb{R}^d\}$, then we are guaranteed to stop when $\mathbb{E}_{x\sim Q_{\xi}}\left[\nabla_{\xi}\log q_{\xi}(x)h(f(x))^{\alpha}\right]=0$, like for a general policy gradient method maximizing $h(f(x))^{\alpha}$ (Papini et al., 2018). Compared to the result for the unconstrained distribution space (Theorem 4.2), we loose the convergence to a fixed point. This property can be recovered under the assumption that the iterate in Equation (5) admits a unique solution for every $P$. In such a case, we will converge to a distribution $Q_{\infty}=\arg\min_{Q\in\mathcal{Q}}\left\{D_{\alpha}(\mathcal{I}_{h\circ f}[Q]\|Q)\right\}$.

---

[3]In Appendix B, we discuss the effects of optimizing a power of $f$ instead of $f$ itself, i.e., when $h=(\cdot)^{\beta}$; while in Appendix C, we discuss cases in which the performance improvement can be obtained for MDPs.

## 5.3 IMPLICIT TRUST REGION

In Theorem 4.3, we have proved that the $\alpha$-Rényi divergence between $\mathcal{I}_{h \circ f}[P]$ and $P$ is bounded. In this section, we answer **(Q3)**, wondering whether similar properties hold when we consider a limited set of distributions $\mathcal{Q} \subseteq \mathscr{P}(\mathcal{X})$. The following result shows that, under a particular form of convexity (van Erven & Harremoës, 2014) of $\mathcal{Q}$, we are able to control the trust region as well.

**Theorem 5.4.** *Let $f \in \mathscr{B}(\mathcal{X}, [0, \infty))$, and $h : [0, \infty) \rightarrow [0, \infty)$ monotonic strictly-increasing. Let $\mathcal{Q} \subseteq \mathscr{P}(\mathcal{X})$ be a $(1 - \alpha)$-convex set (van Erven & Harremoës, 2014, Definition 4), $P \in \mathcal{Q}$, $Q^\dagger \in \arg\min_{Q \in \mathcal{Q}} \{D_\alpha(\mathcal{I}_{h \circ f}[P] \| Q)\}$, and $\alpha \in [0, \infty]$, then it holds that:*
$$D_\alpha \left( Q^\dagger \| P \right) \leqslant D_\alpha \left( \mathcal{I}_{h \circ f}[P] \| P \right) - D_\alpha \left( \mathcal{I}_{h \circ f}[P] \| Q^\dagger \right).$$

Therefore, we are always guaranteed that the trust region induced by $Q^\dagger$ is tighter compared to the one induced by $Q^* = \mathcal{I}_{h \circ f}[P]$ computed in Theorem 4.3, i.e., $D_\alpha \left( Q^\dagger \| P \right) \leqslant D_\alpha \left( \mathcal{I}_{h \circ f}[P] \| P \right)$.

## 6 POLICY OPTIMIZATION VIA OPTIMAL POLICY EVALUATION

In this section, we build a sample-based Off-PL algorithm, named *Policy Optimization via Optimal Policy Evaluation* (PO$^2$PE), which uses Off-VM as an inner loop. For generality, we consider a parametric distribution space $\mathcal{Q}_\Xi = \{Q_\xi \in \mathscr{P}(\mathcal{X}) : \xi \in \Xi \subseteq \mathbb{R}^d\}$, a common setting met in PO.[4]

The structure of PO$^2$PE consists of two nested loops. The outer loop (`Optimization`) acts on the target distribution $q_{\xi_i}$. At the end of each outer iteration $i \in [I]$, the target distribution $q_{\xi_{i+1}}$ is updated with the last behavioral distribution produced by the inner loop $q_{\overline{\xi}_{i,J+1}}$. Instead, the inner loop (`Evaluation`) takes the target distribution provided by the outer loop $q_{\xi_i}$ and provides a new behavioral distribution. At each inner iteration $j \in [J]$, it collects samples $\mathcal{D}_{i,j}$ with the current

---

**Algorithm 1:** PO$^2$PE.

**input** : $\alpha$ divergence order, $h$ function, $f$ function, $\mathcal{Q}_\Xi$ distribution space, $\xi_1$ initial parameter, $n$ batch size

**output:** final parameter $\xi_{I+1} \in \Xi$

1 **for** $i = 1, \dots, I$ **do**    `Optimization`
2     $\overline{\xi}_{i,1} = \xi_i$
3     **for** $j = 1, \dots, J$ **do**    `Evaluation`
4        Collect $\mathcal{D}_{i,j} = \{(x_l, f(x_l))\}_{l \in [n]}$ with $Q_{\overline{\xi}_{i,j}}$
5        Using $(\mathcal{D}_{i,k})_{k \in [j]}$, perform $M$ steps of gradient descent on (Obj).
6     **end**
7     $\xi_{i+1} = \overline{\xi}_{i,J+1}$
8 **end**

---

behavioral distribution $q_{\overline{\xi}_{i,j}}$ and employs them, together with all the samples collected so far $(\mathcal{D}_{i,k})_{k \in [j]}$, to compute the next behavioral distribution $q_{\overline{\xi}_{i,j+1}}$, with the goal of finding the behavioral distribution minimizing the absolute central $\alpha$-moment of the IS estimator (Equation 4). As we shall see, the optimization is performed using samples and by resorting to a penalized objective.

**Sample-based Optimization** The problem of finding the next behavioral distribution parameter $\overline{\xi}_{i,j+1}$ using the samples collected so far $(\mathcal{D}_{i,k})_{k \in [j]}$ is an off-policy learning problem. Let us define $\Phi_{i,j} = \frac{1}{j} \sum_{k \in [j]} q_{\overline{\xi}_{i,k}}$ as the mixture of the $j$ behavioral distributions experienced so far in the inner loop. Instead of directly estimating $D_\alpha(\mathcal{I}_{h \circ f}[Q_{\xi_i}] \| Q_\xi)$, we refer to the (non-central) $\alpha$-moment, which is connected to the original objective through Proposition 5.1. Since we have samples coming from different behavioral distributions, we can use a *multiple* IS estimator Veach & Guibas (1995):

$$\widehat{d}_\alpha \left( \mathcal{I}_{h \circ f}[Q_{\xi_i}] \| Q_\xi; \Phi_{i,j} \right) = \frac{1}{nj} \sum_{k \in [j]} \sum_{l \in [n]} \underbrace{\frac{q_\xi(x_{k,l})}{\Phi_{i,j}(x_{k,l})}}_{(a)} \underbrace{\frac{q_{\xi_i}(x_{k,l})^\alpha}{q_\xi(x_{k,l})^\alpha} h(f(x_{k,l}))^\alpha}_{(b)}. \tag{6}$$

The (a) factor accounts that we are using samples collected with the mixture $\Phi_{i,j}$ to estimate an expectation under $q_\xi$, whereas the (b) factor is the actual variable we want to compute the expectation of, i.e., the $\alpha$-moment. It is simple to prove that the expectation of $\widehat{d}_\alpha$ is indeed the $\alpha$-moment (Papini et al., 2019). To minimize Equation (6), we employ a variance correction to mitigate the effect of finite samples (Metelli et al., 2018), theoretically grounded in the following result.

---

[4]In the action-based PO $\xi = \theta$ are the policy parameters and $\Xi = \Theta$, while in the parameter-based PO $\xi = \rho$ are the hyperpolicy parameters and $\Xi = \mathcal{P}$.

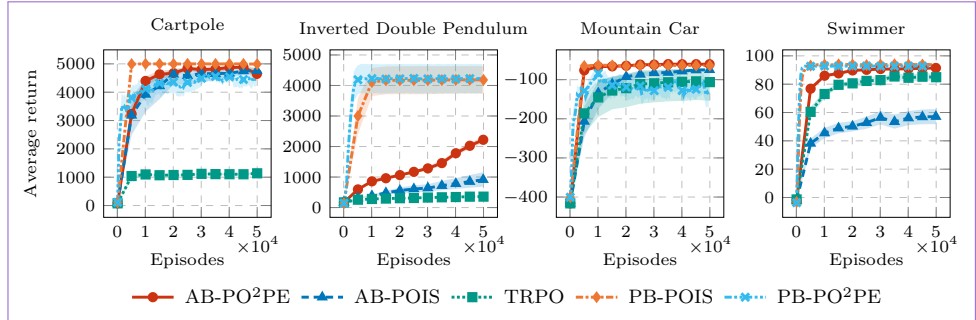

Figure 2: Average return as a function of the number of episodes for different environments and algorithms with batch size $n = 100$, $\alpha = 2$, $h = \mathrm{Id}$, and $J = 1$ (20 runs $\pm$ 95% bootstrapped c.i.).

**Theorem 6.1.** *Let $\mathcal{Q}_\Xi \subseteq \mathscr{P}(\mathcal{X})$ be a set of parametric distributions and let $\xi, \xi_i \in \Xi$. If $\|h \circ f\|_\infty \leqslant \overline{m}$, then, if samples are independent, for every $\delta \in [0,1]$, with probability at least $1 - \delta$ it holds that:*

$$\mathbb{E}_{x \sim \xi}\left[\left(\frac{q_{\xi_i}(x)}{q_\xi(x)} h(f(x))\right)^\alpha\right] \leqslant \underbrace{\widehat{d}_\alpha\left(\mathcal{I}_{h \circ f}[Q_{\xi_i}] \| Q_\xi; \Phi_{i,j}\right) + \overline{m}^\alpha \sqrt{\frac{2 \log \frac{1}{\delta}}{nj} \int_{\mathcal{X}} \frac{q_{\xi_i}(x)^{2\alpha}}{\Phi_{i,j}(x) q_\xi(x)^{2(\alpha-1)}} \mathrm{d}x}}_{(Obj)}.$$

Some remarks are in order. First, the integral within the square root is an upper bound to the variance of the $\alpha$-moment estimator $\widehat{d}_\alpha\left(\mathcal{I}_{h \circ f}[Q_{\xi_i}] \| Q_\xi; \Phi_{i,j}\right)$. In particular, when $\xi = \xi_i$, we obtain the exponentiated Rényi divergence, as illustrated in (Metelli et al., 2020). When all involved distributions are Guassians, it is possible to provide a closed-form tight bound on this quantity (Appendix D). Second, unlike the results available in the literature about concentration of IS estimator, without corrections or transformations, we are able to provide an exponential concentration inequality (dependence on delta of the form $\log(1/\delta)$), instead of a polynomial concentration (dependence of the form $1/\delta$). This is due to the fact that we are dealing with random variables that are bounded to zero from below and they allow applying stronger unilateral Bernstein's concentration inequalities (Boucheron et al., 2009). The reader might object that to optimize the proposed objective function, designed to enforce an implicit trust region, we are actually introducing an additional correction term. This is necessary for theoretical purposes, but, as we shall see in the Section 7, the need for a penalization or constraint is significantly less relevant than in existing approaches, like TRPO (Schulman et al., 2015), or POIS (Metelli et al., 2018).

**Sample Collection** In the action-based setting (AB-PO$^2$PE), we sample $n$ trajectories $\{\tau_l\}_{l \in [n]}$ independently with the policy $\pi_{\overline{\theta}_{i,j}}$ and we build the dataset $\mathcal{D}_{i,j} = \{(\tau_l, \mathcal{R}(\tau_l))\}_{l \in [n]}$. Instead, in the parameter-based setting (PB-PO$^2$PE), we sample independently $n$ policy parameters $\{\theta_l\}_{l \in [n]}$ and for each of them we run policy $\pi_{\theta_l}$ once to generate trajectory $\tau_l$. The corresponding dataset is given by $\mathcal{D}_{i,j} = \{((\theta_l, \tau_l), \mathcal{R}(\tau_l))\}_{l \in [n]}$. For the AB case, the correction in Theorem 6.1 is estimated from samples, as done for the Rényi divergence in (Metelli et al., 2018), since it involves integrals between trajectory distributions, while the closed form exists for Gaussian distributions (Appendix D).

As noted in Section 5, PO corresponds to a constrained setting and, thus, we are in general unable to provide a performance improvement guarantee for every $h$ (Theorem 5.2). We show in Appendix C that performance improvement, independently on $h$, is ensured for deterministic environments and we show some (only theoretical) approaches to extend the guarantee to stochastic environments.

## 7 EXPERIMENTAL EVALUATION

In this section, we provide the experimental evaluation of PO$^2$PE on continuous control tasks. We first compare the learning performance of PO$^2$PE with POIS (Metelli et al., 2018) and TRPO (Schulman et al., 2015) on four benchmarks. Then, deepen two relevant aspects of PO$^2$PE: its robustness to small batch sizes and the effect of the function $h$. All experiments are conducted with Gaussian policies, linear in the state, with fixed variance. The experimental details are reported in Appendix E. **Comparison with POIS and TRPO** In Figure 2, we show the average return as a function of the number of collected episodes, with a batch size $n = 100$, using $\alpha = 2$, $h = \mathrm{Id}$ (identity function), and one inner iteration ($J = 1$). In the Cartpole environment, we observe that the performance of

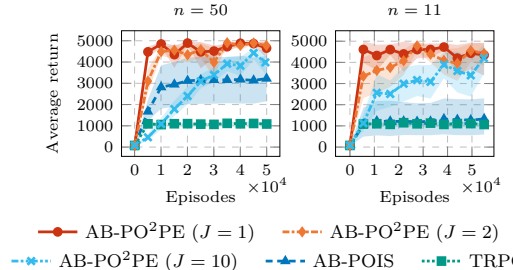

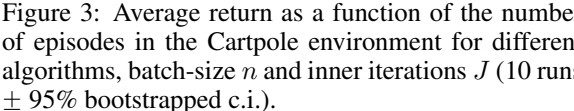

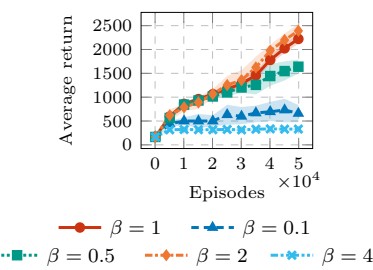

Figure 3: Average return as a function of the number of episodes in the Cartpole environment for different algorithms, batch-size $n$ and inner iterations $J$ (10 runs $\pm$ 95% bootstrapped c.i.).

Figure 4: Average return as a function of the number of episodes in the Inverted Double Pendulum for different choices of $h = (\cdot)^{\beta}$ (5 runs $\pm$ 95% bootstrapped c.i.).

AB-PO$^2$PE is slightly above that of AB-POIS and PB-PO$^2$PE; while the fastest learning curve is shown by PB-POIS. Instead, TRPO converges to a suboptimal policy that fails keeping the pole in the vertical position. In the Inverted Double Pendulum experiment, the gap between AB-PO$^2$PE and AB-POIS and TRPO is more evident. The PB versions outperform the AB ones with PO$^2$PE slightly faster than POIS. In the Mountain Car domain, while AB-POIS, TRPO, and PB-PO$^2$PE display a similar convergence speed, AB-PO$^2$PE and PB-POIS reach the optimal performance faster. Finally, in the Mujoco Swimmer domain (Todorov et al., 2012), AB-PO$^2$PE and TRPO clearly outperform AB-POIS, although the fastest learning curves are displayed by the PB versions of POIS and PO$^2$PE.

**Robustness to Small Batch Sizes**   Based on the previous results, we further investigate the properties of PO$^2$PE in terms of variance control. In the Cartpole domain, we test the robustness to the reduction of the batch size. In Figure 3, we show the average return as a function of the number of collected episodes for batch sizes $n \in \{11, 50\}$ and different number of inner iterations $J$. Also considering the $n = 100$ case (Figure 2), we notice, as expected, that the variance of each setting increases overall as $n$ decreases. Nevertheless, PO$^2$PE proves to be robust, always succeeding in reaching the optimal performance. Differently, POIS suffers the reduced batch size, while TRPO always converging to the same suboptimal policy. The desirable behavior of PO$^2$PE is indeed an effect of the kind of objective function we employ that explicitly accounts for the variance of the estimator, trying to minimize it, and, as we have shown in the previous sections, it allows enforcing an implicit trust region. Finally, a small number of inner iterations $J$ is beneficial for the stability.

**Effect of the Function $h$**   While previous experiments consider $h$ as the identity function, we now investigate the effects of using $h = (\cdot)^{\beta}$, i.e., a power function. In Figure 4, we show the learning curves of the Inverted Double Pendulum for different values of $\beta$. We notice that for $\beta$ close to 1 (0.5, 1, 2) the curves are not very dissimilar, while for too extreme powers (0.1 and 4) the learning performance degrades. This example shows an interesting phenomenon, i.e., even if we optimize a power of return, within certain limits, we are still able to converge to a (near-)optimal policy.

## 8   DISCUSSION AND CONCLUSIONS

In this paper, we have deepened the study of importance sampling beyond its usage as a passive tool for off-policy evaluation and learning. We imported the role of IS as a variance reduction active tool, typical of the Monte Carlo simulation, to the off-policy learning setting. We have illustrated that by minimizing the absolute central $\alpha$-moment of the IS estimator we are able to guarantee the performance improvement for a monotonic transformation of the original objective and eventually converge, at least, to a stationary point. Interestingly, this approach is able to naturally induce a trust region, mitigating the need for an explicit penalization or constraint. The experimental evaluation confirmed our theoretical findings. PO$^2$PE is able to outperform POIS and TRPO on several continuous control tasks. Our algorithm has proved to be remarkably robust to the reduction of the batch size. This is a consequence of using the minimum-variance behavioral distribution that has the benefit of inducing an implicit trust region. We believe that this work contributes to shed light on an appealing facet of off-policy learning with possible new research opportunities. Future works include an extension of the convergence analysis to the sample-based setting and an experimentation of PO$^2$PE coupled with more complex policy architectures.

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

# A PROOFS AND DERIVATIONS

In this appendix, we report the proofs and derivations, we have omitted in the main paper.

## A.1 PROOFS OF SECTION 4

**Proposition 4.1** (Proposition 9 of Ghosh et al. (2020)). *Let $P \in \mathscr{P}(\mathcal{X})$, $f \in \mathscr{B}(\mathcal{X}, [0, \infty))$, and $h \colon [0, \infty) \to [0, \infty)$ monotonic increasing. Then, $\mathcal{I}_{h \circ f}[P]$ is a performance improvement w.r.t. $P$:*

$$\mathbb{E}_{x \sim \mathcal{I}_{h \circ f}[P]}[f(x)] - \mathbb{E}_{x \sim P}[f(x)] = \frac{\mathrm{Cov}_{x \sim P}[h(f(x)), f(x)]}{\mathbb{E}_{x \sim P}[h(f(x))]} \geqslant 0.$$

*Proof.* Let us consider the following derivation:

$$\begin{aligned}
\mathbb{E}_{x \sim \mathcal{I}_{h \circ f}[P]}[f(x)] - \mathbb{E}_{x \sim P}[f(x)] &= \int_{\mathcal{X}} \frac{p(x) h(f(x))}{\mathbb{E}_{x \sim P}[h(f(x))]} f(x) \mathrm{d}x - \mathbb{E}_{x \sim P}[f(x)] \\
&= \frac{\mathbb{E}_{x \sim P}[h(f(x)) f(x)] - \mathbb{E}_{x \sim P}[f(x)] \mathbb{E}_{x \sim P}[h(f(x))]}{\mathbb{E}_{x \sim P}[h(f(x))]} \\
&= \frac{\mathrm{Cov}_{x \sim P}[h(f(x)), f(x)]}{\mathbb{E}_{x \sim P}[h(f(x))]},
\end{aligned}$$

where we have exploited the definition of $\mathcal{I}_{h \circ f}$ and the definition of covariance. The result is obtained by recalling that $h$ is increasing and the covariance between two increasing functions of the same random variable (i.e., $h$ and the identity function) is non-negative (Cuadras, 2002). □

**Theorem 4.2.** *Let $P \in \mathscr{P}(\mathcal{X})$, $f \in \mathscr{B}(\mathcal{X}, [0, \infty))$, and $h \colon [0, \infty) \to [0, \infty)$ monotonic strictly-increasing. Then, the following statements hold:*

*(i) $P$ is a fixed point of $\mathcal{I}_{h \circ f}$, i.e., $\mathcal{I}_{h \circ f}[P] = P$ a.s., if and only if $\mathrm{Var}_{x \sim P}[f(x)] = 0$;*

*(ii) let $\mathcal{X}^* = \arg\max_{x \in \mathrm{supp}(P)}\{f(x)\}$ be the set of maxima of $f$ restricted to the support of $P$. If $\mathcal{X}^*$ is non-empty and measurable then, the repeated application of $\mathcal{I}_{h \circ f}$ converges to a distribution $Q_\infty = \lim_{k \to \infty} (\mathcal{I}_{h \circ f})^k [P]$ with support $\mathcal{X}^*$. In particular $\mathbb{E}_{x \sim Q_\infty}[f(x)] = \max_{x \in \mathrm{supp}(P)}\{f(x)\}$.*

*Proof.* We start with (i). First of all, we observe that since $h$ is monotonically strictly-increasing it holds that $\mathrm{Var}_{x \sim P}[f(x)] = 0$ if and only if $\mathrm{Var}_{x \sim P}[h(f(x))] = 0$. $P$ is a fixed point of $\mathcal{I}_{h \circ f}$, i.e., $P = \mathcal{I}_{h \circ f}[P]$ a.s. if and only if for all $x \in \mathcal{X}$ it holds a.s.:

$$p(x) = \frac{p(x) h(f(x))}{\mathbb{E}_{x \sim P}[h(f(x))]},$$

that occurs if and only if either $p(x) = 0$ ($x \notin \mathrm{supp}(P)$) or $h(f(x)) = \mathbb{E}_{x \sim P}[h(f(x))]$. ($\Rightarrow$) Whenever $p(x)$ is not zero, function $h(f(x))$ is a constant in $\mathrm{supp}(P)$ and, consequently, its variance under $P$ is zero. ($\Leftarrow$) Suppose that $\mathrm{Var}_{x \sim P}[h(f(x))] = 0$, then $h(f(x)) = \mathbb{E}_{x \sim P}[h(f(x))]$ almost surely and, consequently $\frac{p(x) h(f(x))}{\mathbb{E}_{x \sim P}[h(f(x))]} = p(x)$ almost surely. Let us now consider (ii). First of all, we can easily observe that for every $k \in \mathbb{N}$:

$$(\mathcal{I}_{h \circ f})^k [P](x) = \frac{p(x) f(x)^k}{\mathbb{E}_{x \sim P}[f(x)^k]}.$$

Let $f^* = \max_{x \in \mathrm{supp}(P)}\{f(x)\}$, consider the function $g_k(x) = p(x) \left( \frac{f(x)}{f^*} \right)^k$ and the limit:

$$\lim_{k \to \infty} g_k(x) = \lim_{k \to \infty} p(x) \left( \frac{f(x)}{f^*} \right)^k = \begin{cases} p(x) & \text{if } x \in \mathcal{X}^* \\ 0 & \text{otherwise} \end{cases}.$$

Thus, we have:

$$Q_\infty = \lim_{k\to\infty} \left(\mathcal{I}_{h\circ f}\right)^k [P](x) = \lim_{k\to\infty} \frac{p(x)f(x)^k}{\int_{\mathcal{X}} p(x)f(x)^k \mathrm{d}x}$$

$$= \lim_{k\to\infty} \frac{g_k(x)}{\int_{\mathcal{X}} g_k(x)\mathrm{d}x} = \begin{cases} \frac{p(x)}{\int_{\mathcal{X}^*} p(x)\mathrm{d}x} & \text{if } x \in \mathcal{X}^* \\ 0 & \text{otherwise} \end{cases}.$$

Thus, the support of $Q_\infty$ is given by $\mathcal{X}^*$. Consequently, the expectation of $f$ under $Q_\infty$ is given by:

$$\mathbb{E}_{x\sim Q_\infty}[f(x)] = \int_{\mathcal{X}} q_\infty(x)f(x)\mathrm{d}x = f^*.$$

$\square$

**Theorem 4.3.** *Let* $P \in \mathscr{P}(\mathcal{X})$, $f \in \mathscr{B}(\mathcal{X},[0,\infty))$, *and* $h:[0,\infty) \to [0,\infty)$ *monotonic strictly-increasing. Then, for every* $\alpha \in [0,\infty]$, *it holds that:*

$$D_\alpha\left(\mathcal{I}_{h\circ f}[P]\|P\right) = \frac{1}{\alpha-1}\log\frac{\mathbb{E}_{x\sim P}[h(f(x))^\alpha]}{\mathbb{E}_{x\sim P}[h(f(x))]^\alpha}.$$

*In particular, for* $\alpha = 1$ *it holds that:*

$$D_{KL}(\mathcal{I}_{h\circ f}[P]\|P) = \frac{\mathrm{Cov}_{x\sim P}[h(f(x)),\log h(f(x))]}{\mathbb{E}_{x\sim P}[h(f(x))]}.$$

*Proof.* Let us consider the following derivation:

$$J := \int_{\mathcal{X}} \left((I_{h\circ f}[P])(x)\right)^\alpha p(x)^{1-\alpha}\mathrm{d}x = \int_{\mathcal{X}} \left(\frac{p(x)h(f(x))}{\mathbb{E}_{x\sim P}[h(f(x))]}\right)^\alpha p(x)^{1-\alpha}\mathrm{d}x$$

$$= \frac{\mathbb{E}_{x\sim P}[h(f(x))^\alpha]}{\mathbb{E}_{x\sim P}[h(f(x))]^\alpha}.$$

By observing that $D_\alpha\left(I_{h\circ f}[P]\|P\right) = \frac{1}{\alpha-1}\log J$, we obtain the result. For $\alpha = 1$, we provide an independent derivation:

$$D_{\mathrm{KL}}(I_{h\circ f}[P]\|P) = \int_{\mathcal{X}} \frac{p(x)h(f(x))}{\mathbb{E}_{x\sim P}[h(f(x))]} \log\frac{p(x)h(f(x))}{\mathbb{E}_{x\sim P}[h(f(x))]p(x)}\mathrm{d}x$$

$$= \frac{\mathbb{E}_{x\sim P}[h(f(x))\log h(f(x))] - \mathbb{E}_{x\sim P}[h(f(x))]\mathbb{E}_{x\sim P}[\log h(f(x))]}{\mathbb{E}_{x\sim P}[h(f(x))]}$$

$$= \frac{\mathrm{Cov}_{x\sim P}[h(f(x)),\log h(f(x))]}{\mathbb{E}_{x\sim P}[h(f(x))]},$$

where we exploited the definition of covariance in the last line. $\square$

## A.2 PROOFS OF SECTION 5

**Proposition 5.1.** *Let* $P \in \mathscr{P}(\mathcal{X})$, $f \in \mathscr{B}(\mathcal{X},[0,\infty))$, *and* $h:[0,\infty) \to [0,\infty)$ *monotonic strictly-increasing. Then, for any* $\alpha \in (1,\infty)$, *it holds that:*

$$\underbrace{\mathbb{E}_{x\sim Q}\left[\left|\frac{p(x)}{q(x)}h(f(x)) - \mathbb{E}_{x\sim P}[h(f(x))]\right|^\alpha\right]}_{\text{absolute central } \alpha\text{-moment}} \leq \underbrace{\mathbb{E}_{x\sim Q}\left[\left(\frac{p(x)}{q(x)}h(f(x))\right)^\alpha\right]}_{\text{(non-central) } \alpha\text{-moment}} = e^{(\alpha-1)D_\alpha(\mathcal{I}_{h\circ f}[P]\|Q)}\mathbb{E}_{x\sim P}[h(f(x))]^\alpha.$$

*Proof.* First of all, we observe that since $\mathbb{E}_{x\sim Q}\left[\frac{p(x)}{q(x)}h(f(x))\right] = \mathbb{E}_{x\sim P}[h(f(x))]$, for $\alpha \geq 1$, the absolute central $\alpha$-moment is smaller or equal than the (non-central) $\alpha$-moment. Thus, for $\alpha \geq 1$,

we have:

$$
\mathbb{E}_{x \sim Q}\left[\left|\left|\frac{p(x)}{q(x)} h(f(x)) - \mathbb{E}_{x \sim P}[h(f(x))]\right|\right|^{\alpha}\right] \leqslant \mathbb{E}_{x \sim Q}\left[\left(\frac{p(x)}{q(x)} h(f(x))\right)^{\alpha}\right]
$$

$$
= \int_{\mathcal{X}}\left(\frac{p(x) h(f(x))}{\mathbb{E}_{x \sim P}[h(f(x))]}\right)^{\alpha} q(x)^{1-\alpha} \mathrm{d}x \mathbb{E}_{x \sim P}[h(f(x))]^{\alpha}
$$

$$
= \int_{\mathcal{X}}((\mathcal{I}_{h \circ f}[P])(x))^{\alpha} q(x)^{1-\alpha} \mathrm{d}x \mathbb{E}_{x \sim P}[h(f(x))]^{\alpha}
$$

$$
= \exp\left\{(\alpha-1)\frac{1}{\alpha-1} \log \int_{\mathcal{X}}((\mathcal{I}_{h \circ f}[P])(x))^{\alpha} q(x)^{1-\alpha} \mathrm{d}x\right\} \mathbb{E}_{x \sim P}[h(f(x))]^{\alpha}.
$$

By applying the definition of Rényi divergences, we get the result. □

**Theorem 5.2.** *Let $P \in \mathscr{P}(\mathcal{X})$, $f \in \mathscr{B}(\mathcal{X}, [0, \infty))$, and $h : [0, \infty) \to [0, \infty)$ monotonic strictly-increasing. Let $\mathcal{Q} \subseteq \mathscr{P}(\mathcal{X})$, $Q \in \mathcal{Q}$, and $\alpha \in [0, \infty]$, then, it holds that:*

$$
\mathbb{E}_{x \sim Q}[h(f(x))^{\alpha}] - \mathbb{E}_{x \sim P}[h(f(x))^{\alpha}] \geqslant \frac{\mathbb{E}_{x \sim P}[h(f(x))]^{\alpha}}{\alpha-1}\left(e^{(\alpha-1)D_{\alpha}(\mathcal{I}_{h \circ f}[P]\|P)} - e^{(\alpha-1)D_{\alpha}(\mathcal{I}_{h \circ f}[P]\|Q)}\right).
$$

*In particular, for $\alpha = 1$, it holds that (Ghosh et al., 2020, Proposition 6):*

$$
\mathbb{E}_{x \sim Q}[h(f(x))] - \mathbb{E}_{x \sim P}[h(f(x))] \geqslant \mathbb{E}_{x \sim P}[h(f(x))]\left(D_{KL}(\mathcal{I}_{h \circ f}[P]\|P) - D_{KL}(\mathcal{I}_{h \circ f}[P]\|Q)\right).
$$

*Proof.* Let us consider the following derivation:

$$
\mathbb{E}_{x \sim Q}[h(f(x))^{\alpha}] = \int_{\mathcal{X}} q(x) h(f(x))^{\alpha} \mathrm{d}x
$$

$$
= \int_{\mathcal{X}} p(x)\frac{q(x)}{p(x)} h(f(x))^{\alpha} \mathrm{d}x
$$

$$
= \int_{\mathcal{X}} p(x) h(f(x))^{\alpha} \mathrm{d}x + \int_{\mathcal{X}} p(x)\left(\frac{q(x)}{p(x)} - 1\right) h(f(x))^{\alpha} \mathrm{d}x
$$

$$
\geqslant \int_{\mathcal{X}} p(x) h(f(x))^{\alpha} \mathrm{d}x + \frac{1}{\alpha-1}\int_{\mathcal{X}} p(x)\left(1 - \left(\frac{p(x)}{q(x)}\right)^{\alpha-1}\right) h(f(x))^{\alpha} \mathrm{d}x \qquad (7)
$$

$$
= \mathbb{E}_{x \sim P}[h(f(x))^{\alpha}] + \frac{1}{\alpha-1}\int_{\mathcal{X}} p(x) h(f(x))^{\alpha} \mathrm{d}x
$$

$$
- \frac{1}{\alpha-1}\int_{\mathcal{X}} p(x)\left(\frac{p(x)}{q(x)}\right)^{\alpha-1} h(f(x))^{\alpha} \mathrm{d}x
$$

$$
= \mathbb{E}_{x \sim P}[h(f(x))^{\alpha}] + \mathbb{E}_{x \sim P}[h(f(x))]^{\alpha}\frac{1}{\alpha-1}\int_{\mathcal{X}}\left(\frac{p(x)h(f(x))}{\mathbb{E}_{x \sim P}[h(f(x))]}\right)^{\alpha} p(x)^{1-\alpha} \mathrm{d}x
$$

$$
- \mathbb{E}_{x \sim P}[h(f(x))]^{\alpha}\frac{1}{\alpha-1}\int_{\mathcal{X}}\left(\frac{p(x)h(f(x))}{\mathbb{E}_{x \sim P}[h(f(x))]}\right)^{\alpha} q(x)^{1-\alpha} \mathrm{d}x
$$

$$
= \mathbb{E}_{x \sim P}[h(f(x))^{\alpha}]
$$

$$
+ \mathbb{E}_{x \sim P}[h(f(x))]^{\alpha}\frac{1}{\alpha-1}\exp\left\{(\alpha-1)\frac{1}{\alpha-1}\log\int_{\mathcal{X}}\left(\frac{p(x)h(f(x))}{\mathbb{E}_{x \sim P}[h(f(x))]}\right)^{\alpha} p(x)^{1-\alpha} \mathrm{d}x\right\}
$$

$$
- \mathbb{E}_{x \sim P}[h(f(x))]^{\alpha}\frac{1}{\alpha-1}\exp\left\{(\alpha-1)\frac{1}{\alpha-1}\log\int_{\mathcal{X}}\left(\frac{p(x)h(f(x))}{\mathbb{E}_{x \sim P}[h(f(x))]}\right)^{\alpha} q(x)^{1-\alpha} \mathrm{d}x\right\}
$$

$$
= \mathbb{E}_{x \sim P}[h(f(x))^{\alpha}] + \frac{\mathbb{E}_{x \sim P}[h(f(x))]^{\alpha}}{\alpha-1}\left(e^{(\alpha-1)D_{\alpha}(\mathcal{I}_{h \circ f}\|P)} - e^{(\alpha-1)D_{\alpha}(\mathcal{I}_{h \circ f}\|Q)}\right),
$$

where line (7) derived from Lemma A.1. The second inequality was provided in Proposition 6 of (Ghosh et al., 2020). □

**Theorem 5.3.** *Let $P \in \mathscr{P}(\mathcal{X})$, $f \in \mathscr{B}(\mathcal{X}, [0, \infty))$, and $h : [0, \infty) \to [0, \infty)$ monotonic strictly-increasing. Let $\mathcal{Q} \subseteq \mathscr{P}(\mathcal{X})$ and suppose that $h \circ f$ is bounded from above, then, the iterate $Q_{k+1} \in \arg\min_{Q \in \mathcal{Q}}\{D_{\alpha}(\mathcal{I}_{h \circ f}[Q_k]\|Q)\}$ (where possible ties are broken arbitrarily) satisfies:*

(i) *the sequence of divergences $D_\alpha(\mathcal{I}_{h\circ f}[Q_k]\|Q_k)$ is convergent;*

(ii) *the sequence of expectations $\mathbb{E}_{x\sim Q_k}[h(f(x))^\alpha]$ is non-decreasing in $k\in\mathbb{N}$ and converges to a stationary point of $\mathbb{E}_{x\sim Q}[h(f(x))^\alpha]$ w.r.t. $Q\in\mathcal{Q}$.*

*Proof.* Let us consider the sequence of distributions $(Q_k)_{k\in\mathbb{N}}$, generated by the iterate in Equation (5), where possible ties are broken with an arbitrary (possibly with a tie-breaking rule $T_k$ different for every $k$). From Theorem 5.2, we have for every $k\in\mathbb{N}$:

$$\mathbb{E}_{x\sim Q_{k+1}}[h(f(x))^\alpha] - \mathbb{E}_{x\sim Q_k}[h(f(x))^\alpha]$$
$$\geqslant \frac{\mathbb{E}_{x\sim Q_k}[h(f(x))]^\alpha}{\alpha-1}\left(e^{(\alpha-1)D_\alpha(\mathcal{I}_{h\circ f}[Q_k]\|Q_k)} - e^{(\alpha-1)D_\alpha(\mathcal{I}_{h\circ f}[Q_k]\|Q_{k+1})}\right)\geqslant 0,$$

where we simply exploited that $Q_k\in\arg\min_{Q\in\mathcal{Q}}\{D_\alpha(\mathcal{I}_{h\circ f}[Q_k]\|Q)\}$. Thus, $\mathbb{E}_{x\sim Q_k}[h(f(x))^\alpha]$ is a non-decreasing function of $k$. Since $h\circ f$ is bounded, it must be that $\lim_{k\to\infty}\mathbb{E}_{x\sim Q_k}[h(f(x))^\alpha]=\mu_\infty<\infty$, that proves convergence.[5]

Furthermore, being convergent, for $k\to\infty$ it must be that $\mathbb{E}_{x\sim Q_k}[h(f(x))^\alpha]=\mathbb{E}_{x\sim Q_{k+1}}[h(f(x))^\alpha]$ and consequently $D_\alpha(\mathcal{I}_{h\circ f}[Q_k]\|Q_k)=D_\alpha(\mathcal{I}_{h\circ f}[Q_k]\|Q_{k+1})$. Therefore, even if the tie-braking rule prescribes to select $Q_{k+1}\neq Q_k$ we could select $Q_k$ instead, since it lead to the same divergence value. Consequently, being $Q_k$ a solution, we can assert that it is a stationary point of the function $D_\alpha(\mathcal{I}_{h\circ f}[Q_k]\|\cdot)$ (as well as $Q_{k+1}$):

$$0 = \nabla_{q(\cdot)}D_\alpha(\mathcal{I}_{h\circ f}[Q_k]\|Q)|_{Q=Q_k}$$
$$= \frac{1}{(\alpha-1)e^{(\alpha-1)D_\alpha(\mathcal{I}_{h\circ f}[Q_k]\|Q)}\mathbb{E}_{x\sim Q_k}[h(f(x))]}\nabla_{q(\cdot)}\int_\mathcal{X}h(f(x))^\alpha q_k(x)^\alpha q(x)^{1-\alpha}dx|_{Q=Q_k}$$
$$= -\frac{1}{e^{(\alpha-1)D_\alpha(\mathcal{I}_{h\circ f}[Q_k]\|Q)}\mathbb{E}_{x\sim Q_k}[h(f(x))]}\int_\mathcal{X}h(f(x))^\alpha q_k(x)^\alpha q(x)^{-\alpha}dx|_{Q=Q_k}$$
$$= -\frac{1}{e^{(\alpha-1)D_\alpha(\mathcal{I}_{h\circ f}[Q_k]\|Q)}\mathbb{E}_{x\sim Q_k}[h(f(x))]}\int_\mathcal{X}h(f(x))^\alpha dx.$$

We observe that the latter expression is zero if and only if the gradient of $\mathbb{E}_{x\sim Q}[h(f(x))^\alpha]$ w.r.t. $Q$ is zero. Indeed:

$$\nabla_{q(\cdot)}\mathbb{E}_{x\sim Q}[h(f(x))^\alpha] = \int_\mathcal{X}h(f(x))^\alpha dx.$$

Thus, the process converges to a stationary point of $\mathbb{E}_{x\sim Q_k}[h(f(x))^\alpha]$. $\qquad\square$

**Theorem 5.4.** *Let $f\in\mathcal{B}(\mathcal{X},[0,\infty))$, and $h:[0,\infty)\to[0,\infty)$ monotonic strictly-increasing. Let $\mathcal{Q}\subseteq\mathcal{P}(\mathcal{X})$ be a $(1-\alpha)$-convex set (van Erven & Harremoës, 2014, Definition 4), $P\in\mathcal{Q}$, $Q^\dagger\in\arg\min_{Q\in\mathcal{Q}}\{D_\alpha(\mathcal{I}_{h\circ f}[P]\|Q)\}$, and $\alpha\in[0,\infty]$, then it holds that:*

$$D_\alpha\left(Q^\dagger\|P\right)\leqslant D_\alpha\left(\mathcal{I}_{h\circ f}[P]\|P\right) - D_\alpha\left(\mathcal{I}_{h\circ f}[P]\|Q^\dagger\right).$$

*Proof.* The proof is a simple application of Lemma A.2, by taking $Q\leftarrow P$, $Q^*\leftarrow Q^\dagger$, and $P\leftarrow\mathcal{I}_{h\circ f}[P]$. $\qquad\square$

### A.3 Proofs of Section 6

**Theorem 6.1.** *Let $\mathcal{Q}_\Xi\subseteq\mathcal{P}(\mathcal{X})$ be a set of parametric distributions and let $\xi,\xi_i\in\Xi$. If $\|h\circ f\|_\infty\leqslant\overline{m}$, then, if samples are independent, for every $\delta\in[0,1]$, with probability at least $1-\delta$ it holds that:*

$$\mathbb{E}_{x\sim\xi}\left[\left(\frac{q_{\xi_i}(x)}{q_\xi(x)}h(f(x))\right)^\alpha\right] \leqslant \underbrace{\widehat{d}_\alpha\left(\mathcal{I}_{h\circ f}[Q_{\xi_i}]\|Q_\xi;\Phi_{i,j}\right) + \overline{m}^\alpha\sqrt{\frac{2\log\frac{1}{\delta}}{nj}\int_\mathcal{X}\frac{q_{\xi_i}(x)^{2\alpha}}{\Phi_{i,j}(x)q_\xi(x)^{2(\alpha-1)}}dx}}_{(Obj)}.$$

---

[5]Notice that the improvement holds also for $\alpha<1$. Indeed, while it is true that $\frac{\mathbb{E}_{x\sim Q_k}[h(f(x))]^\alpha}{\alpha-1}<0$, but in such a case function $e^{(\alpha-1)(\cdot)}$ is decreasing in its argument.

*Proof.* We start observing that each addendum of $\widehat{d}_\alpha\left(\mathcal{I}_{h\circ f}[Q_{\xi_i}]\|Q_\xi;\Phi_{i,j}\right)$ is non negative. Since all terms are i.i.d., we can apply unilateral Bernstein's inequality (Maurer et al., 2003) that allows achieving an exponential concentration. Thus, for every $\delta\in[0,1]$, with probability at least $1-\delta$ it holds that:

$$\mathbb{E}_{x\sim\xi}\left[\left(\frac{q_{\xi_i}(x)}{q_\xi(x)}h(f(x))\right)^\alpha\right]\leqslant\widehat{d}_\alpha\left(\mathcal{I}_{h\circ f}[Q_{\xi_i}]\|Q_\xi;\Phi_{i,j}\right)$$
$$+\sqrt{2\mathrm{Var}_{x_i\sim\Phi_{i,j}}\left[\widehat{d}_\alpha\left(\mathcal{I}_{h\circ f}[Q_{\xi_i}]\|Q_\xi;\Phi_{i,j}\right)\right]\log\frac{1}{\delta}}.$$

Thus, it remains to provide a bound on the variance term. We exploit the fact that $h(f(x))\leqslant\overline{m}$ and that each addendum represents an i.i.d. random variable:

$$\mathrm{Var}_{x_i\sim\Phi_{i,j}}\left[\widehat{d}_\alpha\left(\mathcal{I}_{h\circ f}[Q_{\xi_i}]\|Q_\xi;\Phi_{i,j}\right)\right]\leqslant$$
$$\frac{1}{(nj)^2}\sum_{k\in[j]}\sum_{l\in[n]}\mathbb{E}_{x_{k,l}\sim\Phi_{i,j}}\left[\left(\frac{q_{\xi_i}(x_{k,l})^\alpha}{\Phi_{i,j}(x_{k,l})q_\xi(x_{k,l})^{\alpha-1}}h(f(x))^\alpha\right)^2\right]$$
$$\leqslant\frac{\overline{m}^{2\alpha}}{(nj)^2}\sum_{k\in[j]}\sum_{l\in[n]}\mathbb{E}_{x_{k,l}\sim\Phi_{i,j}}\left[\left(\frac{q_{\xi_i}(x_{k,l})^\alpha}{\Phi_{i,j}(x_{k,l})q_\xi(x_{k,l})^{\alpha-1}}\right)^2\right]$$
$$=\frac{\overline{m}^{2\alpha}}{nj}\mathbb{E}_{x\sim\Phi_{i,j}}\left[\left(\frac{q_{\xi_i}(x)^\alpha}{\Phi_{i,j}(x)q_\xi(x)^{\alpha-1}}\right)^2\right].$$

$\square$

## A.4 TECHNICAL LEMMAS

**Lemma A.1.** *For every $x\geqslant0$ and $\alpha\in(0,1)\cup(1,\infty)$, it holds that:*

$$x-1\geqslant\frac{1}{\alpha-1}\left(1-\frac{1}{x^{\alpha-1}}\right).$$

*Furthermore, for $\alpha=1$, it holds that:*

$$x-1\geqslant\log x.$$

*Proof.* Consider the auxiliary function $g_\alpha(x)=x-1-\frac{1}{\alpha-1}\left(1-\frac{1}{x^{\alpha-1}}\right)$. We are going to prove that the minimum of $g_\alpha(x)$ is zero. Suppose $\alpha>1$, then $g_\alpha(0)=\infty$ and $g_a(\infty)=\infty$. Thus, the minimum must lie in between and since function $g_\alpha$ is differentiable, we have:

$$\frac{\partial}{\partial x}g_\alpha(x)=1-x^{-\alpha}=0\quad\Longrightarrow\quad x=1.$$

Thus, we have $g_\alpha(1)=0$. Suppose now that $\alpha<1$, we have $g_\alpha(0)=\frac{\alpha}{1-\alpha}>0$ and $g_\alpha(\infty)=\infty$. Thus, again, the minimum must lie in between and with the same calculations as before, we conclude $g_\alpha(1)=0$. The case $\alpha=1$ is trivial. $\square$

**Lemma A.2.** *Let $P\in\mathscr{P}(\mathcal{X})$ and let $\alpha\in(0,\infty)$. Let $\mathcal{Q}\subseteq\mathscr{P}(\mathcal{X})$ be an $(\alpha-1)$-convex (van Erven & Harremoës, 2014, Definition 4) subset of distributions. Let $Q^*\in\mathcal{Q}$ be the $\alpha$-moment projection:*

$$Q^*=\underset{Q\in\mathcal{Q}}{\arg\min}\left\{D_\alpha(P\|Q)\right\}.$$

*If $Q^*$ exists, then for every $Q\in\mathcal{Q}$ if holds that:*

$$D_\alpha(P\|Q)\geqslant D_\alpha(P\|Q^*)+D_\alpha(Q^*\|Q).$$

*Proof.* The proof of the result is inspired to (van Erven & Harremoës, 2014, Theorem 14). Let $\lambda \in [0,1]$ and let us define $Q_\lambda$ as the $(1-\alpha,(1-\lambda,\lambda))$-mixture of $Q^*$ and $Q$:

$$q_\lambda(x) = Z_\lambda^{-1}\left((1-\lambda)q^*(x)^{1-\alpha} + \lambda q(x)^{1-\alpha}\right)^{\frac{1}{1-\alpha}},$$

$$Z_\lambda = \int_{\mathcal{X}} \left((1-\lambda)q^*(x)^{1-\alpha} + \lambda q(x)^{1-\alpha}\right)^{\frac{1}{1-\alpha}} dx.$$

Let us first observe that for $\lambda = 0$, we have $Q_0 = Q^*$ and $Z_0 = \int_{\mathcal{X}} q^*(x) dx = 1$. Since $\mathcal{Q}$ is $(1-\alpha)$-convex and $Q^*$ is the minimizer over $\mathcal{Q}$, it holds that $\frac{\partial}{\partial\lambda} D_\alpha(P\|Q_\lambda)|_{\lambda=0} \geqslant 0$. First of all, we compute:

$$\int_{\mathcal{X}} p(x)^\alpha q_\lambda(x)^{1-\alpha} dx = Z_\lambda^{\alpha-1} \int_{\mathcal{X}} \left[(1-\lambda)p(x)^\alpha q^*(x)^{1-\alpha} + \lambda p(x)^\alpha q(x)^{1-\alpha}\right] dx$$

$$\frac{\partial}{\partial\lambda} Z_\lambda = \frac{1}{1-\alpha} \int_{\mathcal{X}} \left((1-\lambda)q^*(x)^{1-\alpha} + \lambda q(x)^{1-\alpha}\right)^{\frac{\alpha}{1-\alpha}} \left(q(x)^{1-\alpha} - q^*(x)^{1-\alpha}\right) dx.$$

The latter, for $\lambda = 0$, becomes: $\frac{\partial}{\partial\lambda} Z_\lambda\big|_{\lambda=0} = \frac{1}{1-\alpha}\left[\int_{\mathcal{X}} q^*(x)^\alpha q(x)^{1-\alpha} - 1\right]$. For calculation easiness, instead of directly operating on $D_\alpha(P\|Q_\lambda)$, we consider:

$$\frac{\partial}{\partial\lambda} \int_{\mathcal{X}} p(x)^\alpha q_\lambda(x)^{1-\alpha} dx = Z_\lambda^{\alpha-1} \int_{\mathcal{X}} \left[-p(x)^\alpha q^*(x)^{1-\alpha} + p(x)^\alpha q(x)^{1-\alpha}\right] dx,$$

$$+ (\alpha-1)Z_\lambda^{\alpha-2} \frac{\partial}{\partial\lambda} Z_\lambda \int_{\mathcal{X}} \left[(1-\lambda)p(x)^\alpha q^*(x)^{1-\alpha} + \lambda p(x)^\alpha q(x)^{1-\alpha}\right] dx.$$

We now evaluate it at $\lambda = 0$:

$$\frac{\partial}{\partial\lambda} \int_{\mathcal{X}} p(x)^\alpha q_\lambda(x)^{1-\alpha} dx\Big|_{\lambda=0} = -\int_{\mathcal{X}} p(x)^\alpha q^*(x)^{1-\alpha} dx + \int_{\mathcal{X}} p(x)^\alpha q(x)^{1-\alpha} dx$$

$$-\int_{\mathcal{X}} p(x)^\alpha q^*(x)^{1-\alpha} dx \left[\int_{\mathcal{X}} q^*(x)^\alpha q(x)^{1-\alpha} dx - 1\right].$$

For $\alpha \geqslant 1$, we require $\frac{\partial}{\partial\lambda} \int_{\mathcal{X}} p(x)^\alpha q_\lambda(x)^{1-\alpha} dx\big|_{\lambda=0} \geqslant 0$, to obtain:

$$\int_{\mathcal{X}} p(x)^\alpha q(x)^{1-\alpha} dx \geqslant \int_{\mathcal{X}} p(x)^\alpha q^*(x)^{1-\alpha} dx \int_{\mathcal{X}} q^*(x)^\alpha q(x)^{1-\alpha} dx.$$

By applying both sides the log function and dividing by $\frac{1}{\alpha-1} > 0$ we get the result. Symmetrically, for $\alpha < 1$, we require the converse $\frac{\partial}{\partial\lambda} \int_{\mathcal{X}} p(x)^\alpha q_\lambda(x)^{1-\alpha} dx\big|_{\lambda=0} \leqslant 0$. Recalling that $\frac{1}{\alpha-1} < 0$, we obtain the desired result. $\square$

## B  OPTIMIZING MOMENTS OF $f$

In this appendix, we analyze the effect of optimizing a power of $f$ instead of $f$.

**Lemma B.1.** *Let $P \in \mathscr{P}(\mathcal{X})$ and $f \in \mathscr{B}(\mathcal{X}, [\underline{m}, \overline{m}]$. If $\alpha \in (1, \infty)$, it holds that:*

$$0 \leqslant \mathbb{E}_{x\sim P}\left[f(x)^\alpha\right] - \left(\mathbb{E}_{x\sim P}\left[f(x)\right]\right)^\alpha$$

$$\leqslant \frac{\underline{m}^\alpha(\overline{m} - \mathbb{E}_{x\sim P}\left[f(x)\right]) + \overline{m}^\alpha(\mathbb{E}_{x\sim P}\left[f(x)\right] - \underline{m}) - \mathbb{E}_{x\sim P}\left[f(x)\right]^\alpha(\overline{m} - \underline{m})}{\overline{m} - \underline{m}}.$$

*In particular for $\alpha = 2$, we have:*

$$0 \leqslant \mathbb{E}_{x\sim P}\left[f(x)^2\right] - \left(\mathbb{E}_{x\sim P}\left[f(x)\right]\right)^2 \leqslant (\overline{m} - \mathbb{E}_{x\sim P}\left[f(x)\right])(\mathbb{E}_{x\sim P}\left[f(x)\right] - \underline{m}),$$

*that is the Bhatia-Davis inequality for the variance.*

*Proof.* We explicitly consider the optimization problem, for $\alpha \geqslant 1$ and having denoted $\mu = \mathbb{E}_{x \sim P}[f(x)]$:

$$\max_{f: \mathcal{X} \to \mathbb{R}} \int_{\mathcal{X}} p(x) f(x)^{\alpha} \mathrm{d}x$$
$$\text{s.t.} \int_{\mathcal{X}} p(x) f(x) = \mu$$
$$\underline{m} \leqslant f(x) \leqslant \overline{m}.$$

Since $\alpha \geqslant 1$, the optimization problem corresponds to the maximization of a concave function subject to linear and box constraints. It is simple to prove that the optimal solution must assign extreme values to function $f$. Let $p \in [0,1]$, the linear and box constraints enforce:

$$p\underline{m} + (1-p)\overline{m} = \mu \implies p = \frac{\overline{m} - \mu}{\overline{m} - \underline{m}}.$$

From which, by substitution in the objective function, we have:

$$\int_{\mathcal{X}} p(x) f(x)^{\alpha} \mathrm{d}x = p\underline{m}^{\alpha} + (1-p)\overline{m}^{\alpha} = \frac{\underline{m}^{\alpha}(\overline{m} - \mu) + \overline{m}^{\alpha}(\mu - \underline{m})}{\overline{m} - \underline{m}}.$$

$\square$

Thus, in general, optimizing moments of the function $f$, leads to different optimal policies compared to optimizing function $f$ directly. However, from the above results, we see that this discrepancy reduces when the expectation $\mathbb{E}_{x \sim P}[f(x)]$ approaches the extreme value $\overline{m}$ (and also $\underline{m}$, but this is less interesting since we are maximizing). The value $\overline{m}$ can be indeed achieved if we have no restrictions on the distribution space (Section 4).

## C  PERFORMANCE IMPROVEMENT FOR THE MDP SETTING

As already mentioned in Section 5, when we consider the MDP setting, we do not have the full control of the trajectory distribution $p(\tau|\boldsymbol{\theta}) = D_0(s_0) \prod_{t=0}^{H-1} \pi_{\boldsymbol{\theta}}(a_t|s_t) P(s_{t+1}|s_t, a_t)$ as the factors involving the transition model $P$ and the initial state distribution $D_0$ are out of the control of the policy and the policy itself $\pi_{\boldsymbol{\theta}}$ is limited due to the parametrization. As a consequence, performing a step of optimization of the $\alpha$-moment is able to provide improvement guarantees on $J(\boldsymbol{\theta})$ only by choosing the transformation function $h = (\cdot)^{\frac{1}{\alpha}}$ (Theorem 5.2). In this appendix, we prove that under specific conditions we are able to provide guarantees on the improvement of $J(\boldsymbol{\theta})$. Moreover, we show some approaches, with a main theoretical interest, to extend the performance improvement guarantees to the case of stochastic MDPs. They must not to be intended as implementation proposals, but rather as theoretical approaches to cope with this phenomenon.

### C.1  ACTION-BASED PO

We start with the action-based PO setting.

**Deterministic MDPs**  If the MDP is deterministic (i.e., $P$ and $D_0$ are deterministic), we denote the next state as $s_{t+1} = P(s_t, a_t)$. Thus, the trajectory distribution is governed by the policy $\pi_{\boldsymbol{\theta}}$ only: $p(\tau|\boldsymbol{\theta}) = \prod_{t=0}^{H-1} \pi_{\boldsymbol{\theta}}(a_t|s_t)$. In such a case, provided that the policy space $\Theta$ is sufficiently powerful, by minimizing *any* of the $\alpha$-moments, we are able to guarantee the performance improvement. Indeed, in such a case, neglecting the limits of the parametrization $\Theta$, we are in an unconstrained setting.

**Stochastic MDPs**  For general stochastic MDPs, the trajectory density function depends on the transition model probabilities. Thus, we need to design an estimator that get rids of these elements. To this purpose, we denote with $p(\underline{a}|\boldsymbol{\theta})$ as the probability of having observed a sequence of actions $\underline{a} = (a_0, a_1, \ldots, a_{H-1})$ when playing policy $\pi_{\boldsymbol{\theta}}$ in the MDP. If we take $f = \mathbb{E}[\mathcal{R}(\tau)|\underline{a}]$, i.e., the expectation of the return conditioned to the sequence of actions, we are again in an unconstrained

setting, neglecting the limits of $\Theta$. Therefore, we are able to guarantee the performance improvement on $J(\boldsymbol{\theta})$. Nevertheless, this requires to design an estimator of $\mathbb{E}[\mathcal{R}(\tau)|\underline{a}]$, that might be a not easy task.

## C.2 PARAMETER-BASED PO

We now consider the parameter-based PO setting.

**Deterministic MDPs and Policies** In the case of deterministic MDPs and deterministic policy $\pi_{\boldsymbol{\theta}}$, we have that for every $\boldsymbol{\theta} \in \Theta$, the support of the trajectory distribution is made of one trajectory only. In such a case, ignoring the limits of the hyperpolicy parameter space $\mathcal{P}$, we are in an unconstrained setting.

**Stochastic MDPs or Policies** In the case of a stochastic MDP or a stochastic policy (or both), we have a trajectory distribution in which we do not have the possibility to intervene on the policy and transition model factors. Thus, we need to consider an estimators that get rid of these stochastic elements. If we take $f = \mathbb{E}[\mathcal{R}|\boldsymbol{\theta}] = J(\boldsymbol{\theta})$, i.e., the expected return conditioned to a policy parametrization $\boldsymbol{\theta}$, we reduce to the unconstrained setting, with the corresponding performance improvement guarantee. It is worth noting that, compared to the action-based setting, estimating $J(\boldsymbol{\theta})$ is notably simpler compared to the estimation of $\mathbb{E}[\mathcal{R}(\tau)|\underline{a}]$.

## D CLOSED FORM OF THE INTEGRAL FOR GAUSSIANS

In this appendix, we derive a closed form for the integral involved in the computation of the bound of Theorem 6.1 in the case that all involved distributions are Gaussians and for $\alpha = 2$. Let us introduce the notation:

$$\mu = \mathcal{N}(\boldsymbol{\mu_\mu}, \boldsymbol{\Sigma_\mu}), \qquad \phi = \mathcal{N}(\boldsymbol{\mu_\phi}, \boldsymbol{\Sigma_\phi}), \qquad \nu = \mathcal{N}(\boldsymbol{\mu_\nu}, \boldsymbol{\Sigma_\nu}). \tag{8}$$

We have to compute the following integral:

$$\int_{\mathcal{X}} \frac{\mu^4(\mathbf{x})}{\phi(\mathbf{x})\nu(\mathbf{x})^2} d\mathbf{x}.$$

Let us start elaborating on the integrand function, denoting for properly sized vector $\mathbf{x}$ and matrix $\mathbf{S}$, $\|\mathbf{m}\|_{\mathbf{S}} = \mathbf{x}^T \mathbf{S} \mathbf{x}$ and $|\mathbf{S}|$ the determinant of $\mathbf{S}$:

$$\frac{\mu^4(\mathbf{x})}{\phi(\mathbf{x})\nu(\mathbf{x})^2} = \frac{(2\pi)^{-2k}|\boldsymbol{\Sigma_\mu}|^{-2} \exp\left(-2\|\mathbf{x} - \boldsymbol{\mu_\mu}\|^2_{\boldsymbol{\Sigma_\mu}^{-1}}\right)}{(2\pi)^{-k/2}|\boldsymbol{\Sigma_\phi}|^{-1/2} \exp\left(-1/2\|\mathbf{x} - \boldsymbol{\mu_\phi}\|^2_{\boldsymbol{\Sigma_\phi}^{-1}}\right)(2\pi)^{-k}|\boldsymbol{\Sigma_\nu}|^{-1} \exp\left(-\|\mathbf{x} - \boldsymbol{\mu_\nu}\|^2_{\boldsymbol{\Sigma_\nu}^{-1}}\right)}$$

$$= \frac{(2\pi)^{-k/2}|\boldsymbol{\Sigma_\mu}|^{-2}}{|\boldsymbol{\Sigma_\phi}|^{-1/2}|\boldsymbol{\Sigma_\nu}|^{-1}} \exp\left(-2\|\mathbf{x} - \boldsymbol{\mu_\mu}\|^2_{\boldsymbol{\Sigma_\mu}^{-1}} + 1/2\|\mathbf{x} - \boldsymbol{\mu_\phi}\|^2_{\boldsymbol{\Sigma_\phi}^{-1}} + \|\mathbf{x} - \boldsymbol{\mu_\nu}\|^2_{\boldsymbol{\Sigma_\nu}^{-1}}\right).$$

Now, we have to deal with the argument of the exponential:

$$-2\|\mathbf{x} - \boldsymbol{\mu_\mu}\|^2_{\boldsymbol{\Sigma_\mu}^{-1}} + 1/2\|\mathbf{x} - \boldsymbol{\mu_\phi}\|^2_{\boldsymbol{\Sigma_\phi}^{-1}} + \|\mathbf{x} - \boldsymbol{\mu_\nu}\|^2_{\boldsymbol{\Sigma_\nu}^{-1}}$$

$$= -\frac{1}{2}\mathbf{x}^T \underbrace{\left(4\boldsymbol{\Sigma_\mu}^{-1} - \boldsymbol{\Sigma_\phi}^{-1} - 2\boldsymbol{\Sigma_\nu}^{-1}\right)}_{\mathbf{M}} \mathbf{x} + \underbrace{\left(4\boldsymbol{\Sigma_\mu}^{-1}\boldsymbol{\mu_\mu} - \boldsymbol{\Sigma_\phi}^{-1}\boldsymbol{\mu_\phi} - 2\boldsymbol{\Sigma_\nu}^{-1}\boldsymbol{\mu_\nu}\right)^T}_{\mathbf{b}^T} \mathbf{x}$$

$$- \frac{1}{2}\underbrace{\left(4\boldsymbol{\mu_\mu}^T\boldsymbol{\Sigma_\mu}^{-1}\boldsymbol{\mu_\mu} - \boldsymbol{\mu_\phi}^T\boldsymbol{\Sigma_\phi}^{-1}\boldsymbol{\mu_\phi} - 2\boldsymbol{\mu_\nu}^T\boldsymbol{\Sigma_\nu}^{-1}\boldsymbol{\mu_\nu}\right)}_{\mathbf{c}}.$$

We now proceed completing the square:

$$\mathbf{x}^T\mathbf{M}\mathbf{x} - 2\mathbf{b}^T\mathbf{x} = (\mathbf{x} - \mathbf{M}^{-1}\mathbf{b})^T\mathbf{M}(\mathbf{x} - \mathbf{M}^{-1}\mathbf{b}) - \mathbf{b}^T\mathbf{M}^{-1}\mathbf{b}.$$

Thus, we have:

$$-\frac{1}{2}\left(\mathbf{x}^T\mathbf{M}\mathbf{x} - 2\mathbf{b}^T\mathbf{x} + \mathbf{c}\right) = -\frac{1}{2}(\mathbf{x} - \mathbf{M}^{-1}\mathbf{b})^T\mathbf{M}(\mathbf{x} - \mathbf{M}^{-1}\mathbf{b}) + \frac{1}{2}\mathbf{b}^T\mathbf{M}^{-1}\mathbf{b} - \frac{1}{2}\mathbf{c}.$$

Moreover, we observe that the following expression is the density of a $k$-variate normal distribution with mean $M^{-1}b$ and covariance matrix $M^{-1}$:

$$(2\pi)^{-k/2}|\mathbf{M}^{-1}|^{-1/2}\exp\left(-\frac{1}{2}(\mathbf{x}-\mathbf{M}^{-1}\mathbf{x})^T\mathbf{M}(\mathbf{x}-\mathbf{M}^{-1}\mathbf{b})\right).$$

Thus, its integral is 1. Therefore, coming to the initial expression:

$$\int_{\mathcal{X}}\frac{\mu^4(\mathbf{x})}{\phi(\mathbf{x})\nu(\mathbf{x})^2}\mathrm{d}\mathbf{x} = \frac{(2\pi)^{-k/2}|\mathbf{\Sigma_\mu}|^{-2}}{|\mathbf{\Sigma_\phi}|^{-1/2}|\mathbf{\Sigma_\nu}|^{-1}}\left((2\pi)^{-k/2}|\mathbf{M}^{-1}|^{-1/2}\right)^{-1}\exp\left(\frac{1}{2}\mathbf{b}^T\mathbf{M}^{-1}\mathbf{b}-\frac{1}{2}\mathbf{c}\right)$$
$$= \frac{|\mathbf{\Sigma_\phi}|^{1/2}|\mathbf{\Sigma_\nu}|}{|\mathbf{\Sigma_\mu}|^2|\mathbf{M}|^{1/2}}\exp\left(\frac{1}{2}\left(\mathbf{b}^T\mathbf{M}^{-1}\mathbf{b}-\mathbf{c}\right)\right)$$

# E  EXPERIMENTAL DETAILS

In this appendix, we report the experimental details and additional experimental results.

**Infrastructure**   The experiments have been run on two machines:

- 2 x CPUs Intel(R) Xeon(R) CPU E7-8880 v4 @ 2.20GHz (22 cores, 44 thread, 55 MB cache) and 128 GB RAM;
- 4 x Intel(R) Xeon(R) CPU E5-4610 v2 @ 2.30GHz (8 cores, 16 thread, 16 MB cache) and 256 GB RAM.

**Environments**   The environments are the rllab implementations (Duan et al., 2016), MIT license, https://github.com/rll/rllab. The Swimmer environment belongs to the Mujoco suite (Todorov et al., 2012), MuJoCo Personal License, http://www.mujoco.org/.

**Algorithms**   The TRPO implementation is taken from baselines (Dhariwal et al., 2017), MIT licence, https://github.com/openai/baselines. For POIS we use the original implementation (Metelli et al., 2018), MIT license, https://github.com/T3p/baselines.

**Hyperparameters**   In order to properly compare the algorithms, a set of 20 seeds has been chosen. A subset of 5 seeds, underlined, was used to test the performances during the tuning phase. Once the optimal hyperparameters were found, the experiments were extended to the other 15 seeds. In the following, we report the hyperparameter values for PO$^2$PE.

The *shift return* refers to the need for making the return non-negative in order to perform the optimization of the $\alpha$-moment in PO$^2$PE. This procedure is carried out independently at each algorithm iteration by subtracting the minimum return among the ones observed. The *variance init* hyperparameter refers to the logarithm of the standard deviation. All experiments have been carried out with Gaussian policies linear with mean linear in the state variables and constant variance uniform over the state space.

Cartpole

- seeds: 0, 3, 11, 16, 19, 42, 66, 72, 84, 87, 90, 123, 222, 343, 404, 452, 542, 875, 943, 999
- max iters: 500
- policy: linear
- policy init: zeros
- capacity: 1
- inner: 1
- variance init: -1
- step size: 1 / gradient norm
- penalization: True
- delta: 0.75

- max offline iters: 10

Mountain Car

- seeds: 0, 3, 11, 16, 19, 42, 66, 72, 84, 87, 90, 123, 222, 343, 404, 452, 542, 875, 943, 999
- max iters: 500
- policy: linear
- policy init: zeros
- capacity: 1
- inner: 1
- variance init: -1
- step size: 2 / gradient norm
- penalization: True
- delta: 0.9
- max offline iters: 10
- shift return: True

Inverted Double Pendulum

- seeds: 0, 3, 11, 16, 19, 42, 66, 72, 84, 87, 90, 123, 222, 343, 404, 452, 542, 875, 943, 999
- max iters: 500
- policy: linear
- policy init: zeros
- capacity: 1
- inner: 1
- variance init: -1
- step size: 2 / gradient norm
- penalization: True
- delta: 0.99
- max offline iters: 10

Swimmer

- seeds: 0, 3, 11, 16, 19, 42, 66, 72, 84, 87, 90, 123, 222, 343, 404, 452, 542, 875, 943, 999
- max iters: 500
- policy: linear
- policy init: zeros
- capacity: 1
- inner: 1
- log-std init: -0.6
- step size: 1 / gradient norm
- penalization: True
- delta: 0.99
- max offline iters: 10
- shift return: True

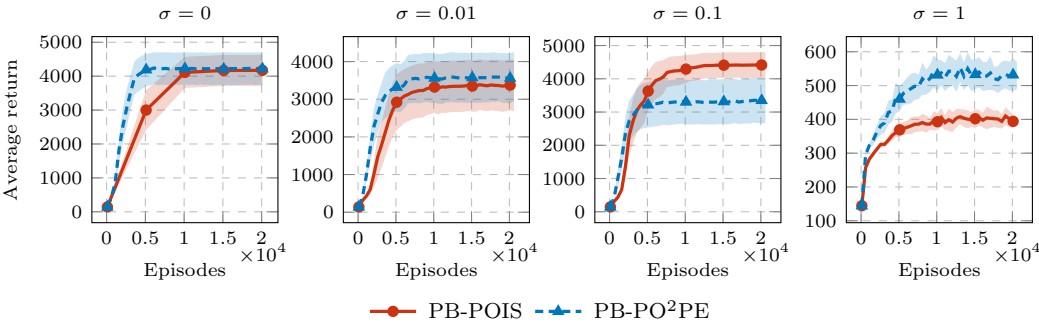

Figure 5: Learning curves comparing PB-POIS and PB-PO$^2$PE with increasing magnitude of the noise (20 runs, 95% c.i.).

For POIS (both AB and PB) and TRPO, the same hyperparameter value have been used, except for the algorithm-specific ones that have been tuned with the same protocol discussed above ($\delta_{KL} \in \{0.001, 0.01, 0.1, 1\}$). In particular, for POIS, we employ the line search procedure presented in the original paper for setting the step-size. The following table summarizes the algorithm-specific hyperparameter values for the different algorithms and environments.

| Environment / Algorithm | PO$^2$PE (delta) | AB-POIS (delta) | TRPO (max kl) |
|---|---|---|---|
| Cartpole | 0.75 | 0.4 | 0.01 |
| Mountain Car | 0.9 | 0.9 | 0.01 |
| Inverted Double Pendulum | 0.99 | 0.1 | 0.001 |
| Swimmer | 0.99 | 0.8 | 0.01 |

| Environment / Algorithm | PB-POIS (delta) | PB-PO$^2$PE (delta) |
|---|---|---|
| Cartpole | 0.4 | 0.6 |
| Mountain Car | 1 | 0.00001 |
| Inverted Double Pendulum | 0.1 | 0.999999 |
| Swimmer | 0.4 | 0.4 |

### E.1 NOISE ROBUSTNESS

As we have shown in Appendix C, using the trajectory return $\mathcal{R}(\tau)$ as function $f$ does no longer allow to provide performance improvement guarantees. Nevertheless, we conjecture that the loss of this property is compensated by the variance reduction implicit in our approach. In the direction of empirically showing this aspect, we tested the parameter-based version of PO$^2$PE in the Inverted Double Pendulum environment, with forced stochasticity in the environment. Specifically, whenever an action is prescribed by the policy the actual action to be executed is obtained by adding while Gaussian noise with standard deviation $\sigma$. The results are shown in Figure 5. We observe that our algorithm is overall competitive with PB-POIS and, in the case of $\sigma = 1$, significantly outperforms PB-POIS.

### E.2 ABOUT RETURN TRANSLATION

Our approach can be employed for non-negative functions $f$. Since in the PO experimental evaluation we employ $f = \mathcal{R}(\tau)$. Under the assumption that the immediate reward is bounded $R(s, a) \in [R_{\min}, R_{\max}]$ for all $(s, a) \in \mathcal{S} \times \mathcal{A}$, we can make the return function with a simple translation and preserving the optimality of policies:

$$\overline{R}(\tau) = \mathcal{R}(\tau) - \underbrace{R_{\min} \frac{1 - \gamma^H}{1 - \gamma}}_{-c_{\min}},$$

where $R_{\min}\frac{1-\gamma^H}{1-\gamma}$ is the minimum achievable return. Of course, we can perform the translation even by using a constant $c \geqslant c_{\min} = -R_{\min}\frac{1-\gamma^H}{1-\gamma}$ and still obtain a translated return that remains positive. It is worth noting, from Theorem 4.3 that the size of the trust region is larger as the constant approaches the its minimum possible value.

For instance, we consider $\alpha = 2$, $f \geqslant 0$ , and we apply a further translation with $c \geqslant 0$. From Theorem 4.3, we have:

$$D_2(I_{+c\circ f}[P]\|P) = \log\frac{\mathbb{E}_{x\sim P}[(f(x)+c)^2]}{\mathbb{E}_{x\sim P}[f(x)+c]^2} = \log\frac{\mathbb{E}_{x\sim P}[f(x)^2]+c^2+2c\mathbb{E}_{x\sim P}[f(x)]}{\mathbb{E}_{x\sim P}[f(x)]^2+c^2+2c\mathbb{E}_{x\sim P}[f(x)]}.$$

Since $\mathbb{E}_{x\sim P}[f(x)^2] \geqslant \mathbb{E}_{x\sim P}[f(x)]^2$, we have that this expression is maximized with the smallest value of $c$, i.e., $c = 0$.

