# OpenReview forum: "Exploiting Minimum-Variance Policy Evaluation for Policy Optimization"
_ICLR.cc/2022/Conference — ICLR 2022 Submitted_

### Official Review · Reviewer_Ruth · 2021-10-30

**Correctness:** 3
**Technical Novelty And Significance:** 3
**Empirical Novelty And Significance:** 2
**Recommendation:** 5
**Confidence:** 3

**Details Of Ethics Concerns:**

No concerns.

**Main Review:**

========== Novelty =============

The idea of relating importance sampling (IS) to policy optimization and policy evaluation is not new. However, this paper digs into the technique of finding variance-minimization behavior distribution of IS, and establishes a few interesting & novel results related to policy optimization.

========== Literature reviews =============

This paper has followed the general logic and narrative of [1], relating "trajectory-based" policy optimization with IS. In "stepwise-based" formulation of RL, which is more common in the general RL literature, [2-5] also adopt ideas of stepwise IS for policy evaluation and optimization. I think such prior work and direct references therein should be of direct interest.

[1] Metelli et al, policy optimization via importance sampling, 2018
[2] Jiang et al, doubly robust off-policy value estimation for reinforcement learning, 2015
[3] Munos et al, safe and sample efficient off-policy reinforcement learning, 2016
[4] Huang et al, from importance sampling to doubly robust policy gradient, 2020
[5] Tang et al, taylor expansion policy optimization, 2020

========== Detailed questions =============

1. Throughout the paper, the authors assume the function f to be non-negative, such that the mathematical treatment of the variance-minimizing behavior distribution is more straightforward. I wonder how this can be implemented in practice, when the return functions are negative. For example, in many high-d continuous control benchmarks (e.g. Ant from openai gym), the initial returns are negative. Since on these environments, the trajectory lengths are not constant, adding a constant positive reward to each time step will not work since it modifies the optimal policy of the problem. If we add a single positive constant to the entire return, what constant should we choose? How does the additive constant impact stability of the algorithm?

2. Eqn 2 shows that the improvement is proportional to the variance of f under the previous distribution. In case the algorithm erroneously converges to a locally optimal deterministic policy, does it mean that the algorithm cannot get out of the local optimal because in such cases the improvement is zero (zero variance from a deterministic policy). Do such observations happen in practice with sampling errors & function approximations?

3. The implicit trust region in Thm 4.3 is interesting from a theoretical perspective, in that one could adjust the effective trust region size based on diff choices of the function h. However, I wonder how useful this result is in practice. In particular, can we choose h in a computationally efficient way to enforce a trust region of particular sizes? How do we estimate the implicit trust region given that it takes the form of a ratio? The trust region also depends on the numerical scale of f itself, which brings back to the issue of tackling problems where f is originally negative -- in this case, adding diff positive offset values will impact trust regions too. Can the authors elaborate on such points?

4. I suggest that the authors update algo 1 to reflect how the algorithm is implemented in practice. From what I understand, algo 1 specifies the ideal algorithm of finding variance minimizing behavior dist at each iteration. However, since the objective is rather diff to estimate, one resorts to an upper bound optimization with finite samples in Eqn 6, backed by Prop 5.1. Then the authors introduce a finite sample correction in Thm 6.1. This makes the algorithm a bit difficult to understand.

5. On experiments, I think the TRPO results in Fig 2-4 are a bit questionable. TRPO has long been considered a good performing algorithm if not state-of-the-art, yet I am rather surprised to see TRPO gets underperformed by many IS baseline algorithms on such low-dimensional problems. I wonder if the TRPO implementation deviates much from the original TRPO algorithm in Schulman et al, 2015.
Do the authors adopt a trajectory-based TRPO implementation or the stepwise-based TRPO implementations in Schulman et al, 2015?

6. Trajectory-based formulation of policy optimization is easier for mathematical treatment but I would argue it has limited application in large-scale practices. I think this is because product of IS ratios usually induce too much variance that in practice such methods are not quite stable. On toy examples, I could understand that such methods are still useful. This leads me to wonder -- is there a way to extend such formulations to step-wise optimization algorithms (PPO, TRPO, IMPALA, MPO...), in which case such ideas could be made more practically appealing.

7. All toy examples seem to have a natural non-negative return function. I wonder if the algorithm could be useful when returns are by design negative, in which case the choice of h is critical.

8. Minor note: I suggest replacing Q_theta by another notation as Q_theta confuses with Q-function, whereas here it refers to the parametric dist.

**Summary Of The Paper:**

The paper analyzes the connection between searching for an optimal behavior policy that minimizes variance, and policy improvement in RL. The paper shows a number of interesting theoretical results, such as non-negative policy improvements, convergence and connections to implicit trust regions between consecutive policy iterates. The paper also shows results in constrained case where the policy class is parameterized. Finally, the paper shows empirical results on a few low-dimensional RL examples.

**Summary Of The Review:**

Overall, I think the paper is solid in its theoretical contributions, yet is a bit lacking in a few important practical aspects.

1. Trajectory-based formulations of policy optimization is not practical when horizons are long. The application of such techniques to RL feels very black-box, as they do not leverage the Markov structure of the problem at all. I think extending such techniques to step-wise based policy optimization algorithm would make the approach more practically appealing.

2. Empirical results are a little bit suspicious in that TRPO underperforms in a significant manner. I would even argue that the combination of this approach with ideas from TRPO can make a more solid empirical case.

---

> ### Author Response · Authors · 2021-11-20
> **Reply to Reviewer Ruth**
>
>
> We thank the reviewer for the insightful comments and suggestions and for having appreciated the novelty of our results.
>
> 1. We agree with the reviewer that in episodic environments translating up each immediate reward by a constant introduces a bias, unless the reward of the absorbing state is translated as well. In our approach, we add a constant to the return function instead of to the immediate rewards. This constant is the opposite of the minimum theoretical return that can be experienced in the environment. Assuming that in the original environment $r(s,a) \in [R_{\min}, R_{\max}]$, the minimum return is $\frac{R_{\min}}{1-\gamma}$ in the infinite-horizon discounted setting and $R_{\min} H$ in the finite-horizon (horizon $H$) undiscounted setting. Acting directly on the return does not change the optimal policy and requires the knowledge of $R_{\min}$ only.
>
> 2. The reviewer is correct: any deterministic policy is a local optimum of our optimization problem, just as it happens in the traditional policy gradient. We have not experienced such phenomena in practice.
>
> 3. The main interest, from a practical perspective, about the trust-region result of Theorem 2.3 is that it guarantees the existence of such a trust-region for every order $\alpha$ of the Renyi divergence and this property is not guaranteed by the majority of the trust-region methods. Nevertheless, computing or estimating the size of such a region is hard since, as the reviewer suggested, because of the presence of the function $f$ itself in the expression of the trust region. In practice, we regard $h$ as a hyperparameter of the algorithm. Furthermore, the relation between the translation of function $f$ and the size of the trust region is interesting. Indeed, the larger the translation constant the smaller the trust-region (this is visible in Theorem 4.3). We will add this observation in the revision.
>
> 4. We will update the algorithm in order to reflect the actual implementation and, in particular, the gradient-based optimization of the objective function in Theorem 6.1.
>
> 5. The TRPO implementation is that of Open AI Baselines (https://github.com/openai/baselines) with the only modification that the sampling procedure employs the number of episodes instead of the number of transitions to define the batch size (in order to get a fair comparison with POIS and our algorithm). Nevertheless, nothing is changed concerning the objective function of TRPO. The learning curves of TRPO we are showing are in line with the results presented in [2].
>
> [2] Metelli, Alberto Maria, Matteo Papini, Nico Montali, and Marcello Restelli. "Importance Sampling Techniques for Policy Optimization." J. Mach. Learn. Res. 21 (2020): 141-1.
>
> 6. We agree with the reviewer that the trajectory-based formulation has the drawback of considering the product of the policy ratio which might induce a large variance. We think this is an even stronger motivation for exploring variance reduction approaches such as the one we have presented in our paper. Nevertheless, we think that this idea could be extended to the step-based setting by considering the search of the policy that minimizes the variance of the objective function with the critic. This could be an interesting direction for future research.
>
> 7. See Answer to Question 1
>
> 8. We will change the symbol in the revision

---

> > ### Comment · Reviewer_Ruth · 2021-11-28
> > **Reply to your rebuttal**
> >
> > Thanks a lot to the authors for the clarifications.
> >
> > 1. Adding a constant to each trajectory. I think this makes much more sense compared to adding constant rewards to each time step. However, I wonder if we usually assume access to $R_{min}$ in the first place, or do we dynamically adjust this constant? Also, it also seems that in addition to $R_{min}$ itself, any constant in the form of $R_{min}+b,b>0$ is a valid constant to add. Since each $b$ can lead to different algorithmic behavior, my understanding is that $b$ is also a meaningful hyper-parameter to disucss?
> >
> > 2. Trust region. Thanks for your reply on this point. I agree that the existence of such a trust region is interesting from a theoretical standpoint. However, to complete the discussion, I think it'd be interesting to see whether e.g. enforcing a particular value of the trust region actually stabilizes the performance. To my understanding, this is why many trust region algorithms tend to be more stable.
> >
> > 3. TRPO implementation: I agree that the results in the current paper are consistent with those in [1]. I have a few comments:
> >
> > (1) In [1], they only compare TRPO/PPO on small scale problems, which makes it hard to verify with TRPO/PPO results in the original paper.
> >
> > (2) We can verify the result of Swimmer in [1] and the PPO paper [2]: in [1], TRPO achieves 96 on average (Table 6) whereas in [2] TRPO achieves 120 (in fact outperform PPO, among other baselines). It is not quite clear to me why TRPO's performance in [1] differs from [2], since this might imply that the implementation of TRPO used in [1] is quite sub-optimal. While I understand that in practice researchers use different code base for comparison, I think it is reasonable to refer to the performance of the original paper [2].
> >
> > (3) TRPO is a step-based algorithm while the methods in this paper are trajectory-based. Overall, my impression is that step-based algorithm should be more sample efficient than trajectory-based algorithms, when properly tuned. This, however, does not contradict with the contribution of this paper. I think it'd be more interesting if we can compare the current method with a similar trajectory-based algorithm with explicit trust region constraint (constraints on the trajectory distribution), and see if it performs better. This removes the 'step-based' vs. 'traj-based' discrepancy as a confounding factor.
> >
> >
> > [1] Importance Sampling Techniques for Policy Optimization.
> > [2] Proximal policy optimization

---

> > > ### Author Response · Authors · 2021-11-29
> > > **Re: Reply to your rebuttal**
> > >
> > > Thank you for the reply.
> > >
> > > 1. In the implementation of the algorithm, $R_{\min}$ is progressively updated as the algorithm proceeds. The reviewer is correct: adding any positive constant $b$ to $R_{\min}$ leads to a valid translation. In principle, $b$ could be treated as a hyperparameter, but we think that a reasonable choice for $b$ is the value leading to the largest trust region possible. In the revision, we have shown in Appendix E.2 that $b=0$ is such a value.
> > >
> > > 2. We understand the reviewer's point. On one hand, the fact that our approach enforces a trust region whose "value" cannot be tuned might be seen as a limitation. On the other hand, classical trust-region methods, in which such a "value" is externally set, introduce this additional hyperparameter whose optimal value is likely problem-dependent.
> > >
> > > 3.
> > >
> > > (1) We agree on this. Our experimental setting is different from the one employed in the original TRPO/PPO papers.
> > >
> > > (2) The main differences between our setting and the one of [2] is that, similarly to [1], we are using the **rllab** implementations of the environments (as specified in Appendix E), whereas in [2] the **OpenAI Gym** implementations are employed (as specified in Section 6.1). Moreover, the policy models are different. Indeed, we are using linear policies, [1] in Table 6 is using a three-layer MLP with [100, 50, 25] neurons (tanh activation), and [2] is using a two-later MLP with 64 neurons each (tanh activation).
> > >
> > > (3) We understand the reviewer's point. To the best of our knowledge, the trajectory-based algorithm that is closer to a trust region method is POIS, although it does not employ an explicit constraint but a penalization of the objective function (in any case based on a divergence between trajectory distribution).

---

### Official Review · Reviewer_9jPT · 2021-11-02

**Correctness:** 4
**Technical Novelty And Significance:** 3
**Empirical Novelty And Significance:** 3
**Recommendation:** 10
**Confidence:** 4

**Main Review:**

Strengths:

The paper considers an important problem: the minimization of variance in off-policy policy gradient methods.
Usual (unbiased) estimators rely on importance sampling. In the community, such estimators, are considered to suffer from high variance.
The high variance comes from the "mismatch" between the behavior policy and the optimization policy.
Cleverly, the authors turn the utilization of importance sampling as a variance minimization approach, by finding the optimal behavior policy w.r.t. this objective.

The method is sound and sheds light on different aspects of of-policy gradient estimation. In particular, this minimum-variance approach is shown to theoretically improve the policy and to provide implicitly a trust region (which is usually needed in off-policy policy optimization).

The authors provide a theoretical analysis of their approach, considering approximation due to the impossibility of the realization of the trajectory distribution.

The authors provide also an empirical analysis of their algorithm, comparing it with TRPO and POIS, analyzing its robustness w.r.t. small batch-sizes (which, usually, can affect negatively the variance), and analyzing the effect of the function $h$.

To conclude, the problem considered is important, the proposed method is sound, the theoretical and empirical analysis are satisfying.
In addition, I think that the authors made a good job in drawing connections with related work.

WEAKNESSES

As also considered in POIS, the usual policy gradient can happen at two different levels: parameter and action. In fact, the authors of POIS provided both a parameter-based POIS and an action-based POIS.

The most common policy gradient methods, like SAC, PPO, TRPO, DDPG, are action-based. It seems to me that PO^2PE, instead, is parameter-based. I ask the author to confirm or deny this.

I think that this is actually the most unclear point of the paper. All the derivations are kept abstract, by considering generic samples $x$. It is not clear to me if the sample $x$ (for example in Equation 6) are policy parameters (i.e., like in $\omega_{\rho'/\rho}(\theta)$ in page 16 of https://www.jmlr.org/papers/volume21/20-124/20-124.pdf) are trajectories, or are something different. In my understanding, $x$ should be regarded as "lower-level" policy parameters, and $q$ is a process sampling policy parameter (what "A Survey on Policy Search for Robotics", calls Exploration in Parameter Space, page 16).

If this is the case, then I do not fully understand the comparison with TRPO. TRPO performs exploration in the action space.

Further, I think it was not specified in parameter-base POIS or action-based POIS was utilized.

My suggestion, however, is to well clarify what $x$ is at the beginning of the document.

In general, I think that while the authors did a great job in explaining the theory, they used little space in the paper to describe the actual algorithm. In general, I would enjoy more reading the paper, if more and clear references were made in the theory about the RL problem and the designing of the algorithm.


To outline another minor issue, the experiments consider only low-dimensional benchmarks. The usage of a more complicated high-dimensional environment would have made this submission stronger.






**Summary Of The Paper:**

Importance sampling (IS) is at the core of many off-policy reinforcement learning (RL) algorithms.
The common use of IS in RL is to consider a _fixed_ policy behavior, say $q$ and an optimization policy $p$. The samples are acquired with the behavior policy $q$. The objective is to maximize

$$
\max_p \mathbb{E}_{x\sim q}\left[\frac{p(x)}{q(x)}f(x)\right].
$$

This approach suffers from high variance.

This paper highlight that originally, IS was thought to _reduce_ the variance of Monte-Carlo estimates. The main idea of this paper is to use IS as a variance minimization technique. It is interesting to see that the $\min_q Var[p(x)/q(x)f(x)]$ leads to a _policy improvement_.

The paper is structured as follow:

1. The authors show in Equation 1 the _minimum variance_ $q^*$. This analytical result is (as far as I understand) purely theoretical and cannot be used directly in RL. Note that Equation 2 also shows how the minimum variance behavior is a policy improvement.
The authors, therefore, suggest that one could simply iterate the minimum variance to consistently improve the policy. At this point, a few questions remain open: will this process converge to a local or global optimum? Can one quantify the divergence between consecutive distributions?

2. To study the convergence properties, the authors introduce an operator $\mathcal{I}$ which takes in input a distribution $p$ and produces the next _minimum variance_ distribution $q$, which will be the next behavior policy. The operator $\mathcal{I}$ has several fixed points: if $p$ is deterministic then $p = \mathcal{I}p$. Theorem 4.2 ensures that the iterated application of $\mathcal{I}$ converges to the optimum in the domain of $p$.

3. Often, trust-region methods are used in RL to ensure stability in the learning process. Theorem 4.3 shows that the application of $\mathcal{I}$ produces a policy with bounded Renyi divergence with the previous policy. Interesting, this result is valid for any order of the divergence. This result is stronger than many other results in RL, where the divergence is only bounded for a unique order (typically 2: KL divergence), leaving the divergence on other orders unbounded.

4. However, the finding $q^*$ is often not possible. In short, if we consider a generic set of distribution, the application of $\mathcal{I}$ to an element of the set, might produce a distribution not contained in the set. The authors need at this point to devise an _approximation_ of $\mathcal{I}$, and check again for the convergence properties.
The first approximation consists in minimizing a $\alpha$-Renyi divergence. The difference between classic divergence minimization (or constraint) approaches is that in this paper, the authors are constraining divergence between "return distribution" (even though modified by the monotonic function $h$) rather than the divergence between parameter distribution or state distribution like in REPS or TRPO (@authors: correct me if I am wrong on this point).

5. The authors analyze the policy improvement of this "approximated" operator. They notice that the policy improvement happens only for some particular choice of $h$

6. The authors prove in Theorem 5.4 that the approximated operator has a tighter trust region (for every single choice of $\alpha$) w.r.t. the unconstrained operator $\mathcal{i}$.

7. The authors provide a practical algorithm in Section 6, and evaluate its efficacy in four classic control benchmarks. They also provide an analysis of the hyperparameters used (monotonic function $h$ and batch-size).


To summarize, the paper proposes a different use of IS, which allows both for variance reduction and policy improvement.
The improvement scheme is new (although it can be related to other approaches) and sheds new light on policy gradient optimization.





**Summary Of The Review:**

PROS

The paper considers an important problem and proposes a sound solution to it. I did personally enjoy learning the application of the importance sampling technique to reduce the estimation variance. I think that the theory shed new light on off-policy policy gradient algorithms.

The theoretical and empirical analyses are satisfying.

CONS

The clarity of exposition could be improved by making the theory more linked to the actual RL problem.

The authors considered only low-dimensional problems in the empirical evaluation, raising the question of whether the algorithm could be applied to more complex domains. However, I think that the theoretical analysis compensates for this minor weakness.


I would like the authors to clarify the doubt I wrote in the main review.

---

> ### Author Response · Authors · 2021-11-20
> **Reply to Reviewer 9jPT**
>
> We thank the reviewer for the positive feedback. We are happy that the reviewer understood that our main goal with this paper is to shed light on a view of importance sampling that is typically disregarded in the reinforcement learning literature. We agree that the passage between the general formulation and the actual algorithm is quite abrupt. We are working on a revision of the paper to fix this, anticipating, as the reviewer suggests, the meaning of $x$ in the policy optimization setting. We will submit it as soon as possible.
>
> Our general approach applies to both action-based and parameter-based policy optimization. The experiments currently shown in the paper are in the action-based setting for both our PO2PE algorithm and for POIS. As for completeness, also based on the other reviewers’ comments, we are running experiments in the parameter-based setting that we will include in the revision.

---

> > ### Comment · Reviewer_9jPT · 2021-11-23
> > **Answer**
> >
> > Thanks for your answer. I appreciate the changes made in the paper, especially the distinction between the two algorithms Action-Based and Parameter-Based, and the new experiments. I think that the clarity of the paper benefited from these changes.

---

### Official Review · Reviewer_pKkx · 2021-11-02

**Correctness:** 3
**Technical Novelty And Significance:** 3
**Empirical Novelty And Significance:** 1
**Recommendation:** 3
**Confidence:** 4

**Main Review:**

## Strengths:

1. The paper raises an interesting connection between the two types of importance sampling. Usually in RL we are thinking of using importance sampling to re-weight off-policy updates, but here the authors take inspiration from the Monte Carlo estimation community and instead try to find the policy that will yield the lowest variance importance-weighted estimator. While this connection seems to have been briefly made by Hanna et al. for policy evaluation (as cited in the paper), the idea to use this in policy optimization seems novel and interesting.
2. The connection yields a novel algorithm that will potentially have different strengths than previous policy optimization methods. This has the potential to expand the array of available policy optimization algorithms.



## Weaknesses:

1. Clarity. The entire paper is presented at such a level of generality that it is often unclear how the method and theory pertain to the RL problem and how the algorithm compares with other methods. One particularly stark example of this problem is the function $ f$. Throughout the paper everything is presented as trying to optimize $ f$, but it is never made clear how $ f$ relates to the RL problem in particular (which seems to be the ultimate goal of the paper). This makes it especially difficult to compare POPE to related algorithms. There is not a clear explanation in the paper about how POPE is similar and different to TRPO for example in a formal way. This sort of context would make the paper much easier to understand.
2. A fundamental problem is hidden by the lack of clarity. The issue of defining $ f $ is finally addressed in Appendix C where it is explained that $ f $ can only easily be defined for RL problems in deterministic MDPs and even then relies on having a "sufficiently powerful" policy space, where the exact definition of this sufficiency is left vague. The fact that the algorithm cannot function in stochastic environments without estimate $ \mathbb{E}[\mathcal{R}(\tau)|\underline{a}]$ (according to appendix C) seems like a serious problem.  This seems to be a fundamental limitation of the proposed algorithm and the misdirection and obfuscation of the problem in the paper is a serious issue. Moreover, it is very unclear how the algorithm is actually implemented in experiments given this limitation.
3. The paper provides no clear reason to prefer the proposed algorithm over alternatives. There doesn't seem to be any theory as to what particular cases the proposed algorithm would improve over something like TRPO. There is some evidence in the experiment section of a slight improvement on some small-scale problems, but I will address this more fully below. In general, the paper lacks a coherent comparison to related policy optimization methods.
4. The algorithm is proposed as a multi-level optimization where evaluation is carried out inside of an inner loop. However, in practice the number of iterations of the inner loop is set to $ J=1$. This calls into question why the algorithm needed the complicated multi-level structure to begin with. Is the inner loop of the algorithm necessary? Why might it be beneficial? When $ J=1$, the algorithm seems to be much simpler. Is it more similar to related policy optimization algorithms in this case?
5. I see a few problems with the experiments in the paper:
   1. Hyperparameter tuning. Were all algorithms tuned in the same way? Appendix E does not give enough information to tell. It seems that 4 values of one hyper parameter were tuned for TRPO, but that perhaps several times more hyperparameters were tuned for POPE (between delta, std/variance init, and step size, not to mention $ \beta$ and $J$ which also seem to be tuned from Figures 3 and 4).
   2. Scale of experiments is small. A major motivation for TRPO and PPO is that they work well wit neural net policies on large-scale problems. It is difficult to conclude that POPE is a better or more useful algorithm without a comparison in those larger settings with neural network policies and larger MDPs.
   3. The batch-size experiments don't seem to have very clear motivation. The paper emphasizes some experiments using very small batch sizes, but it is not clear why the different between batch sizes of ~10 and ~50 matters very much. They are all small enough to easily fit into memory and be computed quickly, especially with linear policies. It's unclear why this is a benefit of the proposed approach.



**Typos**:

- Page 4 after Thm 4.2: "all three properties", but there are only two properties listed in the theorem
- Page 5, example 4.1, last sentence: "elad" should be "lead"
- Page 5, section 5, second paragraph "More in general, when consider" is not grammatical

**Summary Of The Paper:**

This paper proposes a novel policy optimization algorithm called POPE that is based off of ideas about how to use importance sampling for variance reduction from the Monte Carlo estimation community. The paper proves that the policy that minimizes the variance of the evaluation step actually provides a policy improvement, and then creates an algorithm that repeatedly estimates the minimal variance policy as a policy improvement step. It concludes with some small-scale experiments with linear policy classes on low-dimensional control tasks and claims modest improvements over TRPO and POIS baselines.

**Summary Of The Review:**

While the paper provides an interesting connection that yields a (to my knowledge) novel algorithm, I don't think it is ready for publication at this time. There are major issues with clarity that make it unclear how the algorithm is implemented at all and I am wary of a few parts of the experimental setup too. But, the core idea is interesting enough that I would encourage the authors to resolve these issues and re-submit the paper to another conference.

---

> ### Author Response · Authors · 2021-11-20
> **Reply to Reviewer pKkx (II)**
>
> 3. We believe we have shown at least three reasons why the approach we propose could be preferred in practice. Besides outperforming POIS and TRPO in the experiments of Figure 2, our approach: (i) ensures performance improvement at least for deterministic environments; (ii) provides a trust-region guaranteed on Renyi divergences of *any* order; (iii) is significantly robust to batch size reduction (Figure 3). While being aware that the proposed algorithm has not been tested on higher dimensional environments and coped with more complex policies (e.g., neural networks), we believe that these three aspects provide a solid basis for a view of importance sampling that has not been explored yet. We think that the proposal of a practical approach based on this view of importance sampling and the corresponding evaluation of more complex environments and policies is beyond the scope of this paper.
>
> 4. The number of inner iterations $J$ was selected equal to $1$ because, empirically, this was the best value (as also shown in Figure 3). While it is true that for $J=1$ there is only one inner iteration, we point out that the bi-level structure of the whole algorithm is preserved. Indeed, *each* inner iteration corresponds, in principle, to the solution of a full offline optimization problem (i.e., the minimization of the objective function in Theorem 6.1). In practice, full optimization is replaced with a sequence of gradient updates.
>
> 5.1. All algorithms have been tuned in the same way based on the best performance obtained over the first 5 seeds. The hyperparameters of PO2PE that have been tuned are $J$, $\delta$, the step size, the variance initialization, while the others have been kept fixed. For TRPO and POIS the step size and the variance initialization have been tuned. We point out that the curves of these latter two algorithms appear in line with the ones presented in [2].
>
> [2] Metelli, Alberto Maria, Matteo Papini, Nico Montali, and Marcello Restelli. "Importance Sampling Techniques for Policy Optimization." J. Mach. Learn. Res. 21 (2020): 141-1.
>
>
> 5.2. See Answer to Question 3.
>
> 5.3. We are convinced that the robustness to small batch sizes is one of the most remarkable advantages of our approach. Indeed, when looking at the batch size, we are not concerned about possible memory issues when considering large batch sizes, rather about the large variance that is injected when considering small batch sizes, leading to poor performance. Our experimental evaluation showed that PO2PE exhibits good learning curves even in the presence of small batch sizes when the other baselines show significant performance degradation.
>
> ***Typos***
> Thank you for reporting the typos; we will fix them.

---

> > ### Comment · Reviewer_pKkx · 2021-11-26
> > **post rebuttal comment**
> >
> > Thanks for the response. After reading it, I think that the main issues I initially raised have not been resolved.
> >
> > In particular, I think there is still a major issue with clarity and obfuscation. While the paper is introduced and framed as a strategy for RL, most of the presentation focuses on an "unconstrained" setting where the distribution over trajectories is not constrained by the MDP. In addition to be a very confusing way to present the algorithm, this obscures some fundamental issues with the approach that are not at all discussed outside of an appendix. Specifically, the improvement guarantee seems to only hold in deterministic environments unless we have a good estimate of the expected return conditioned on a trajectory of actions. This is briefly addressed only in Appendix C and seems to be attempted to be hidden in the main text and especially in the framing of the paper in the abstract, intro, and setup.
> >
> > Second, I am not totally convinced by the empirical evaluation for a few reasons. First, in the main experiment (Figure 2), the proposed algorithm seems to be outperformed by the baseline PB-POIS. Second, it seems the authors tune twice as many hyperparameters for their algorithm compared to the baselines. Third, all the experiments are only done on small environments with linear models. I recognize that the main contribution of the paper is perhaps not to scale up the algorithm, but this is still a clear weakness of the evaluation to me.
> >
> > I see that perhaps if we restrict to action-based algorithms the improvement of POPE over the baseline is more clear, but am not sure how this restriction would be justified.  The robustness to batch size is potentially interesting if it indeed shows lower variance of the gradient, but I think it is still a bit unclear as presented because the smaller batch sizes are not actually leading to dramatically improved sample complexity in the main experiments (perhaps because there is no comparison to PB-POIS in the batch size experiments).
> >
> > Let me know if there is anything in particular that you think I am mistaken about.

---

> > > ### Author Response · Authors · 2021-11-27
> > > **Re: post rebuttal comment (I)**
> > >
> > > We thank the reviewer for the answer. We think that the rebuttal revision has substantially solved the issues initially raised by the reviewer. Nevertheless, we commit to further improving the clarity of the paper.
> > >
> > > >**In particular, I think there is still a major issue with clarity and obfuscation.**
> > >
> > > >**This is briefly addressed only in Appendix C and seems to be attempted to be hidden in the main text and especially in the framing of the paper in the abstract, intro, and setup.**
> > >
> > > We firmly disagree with the reviewer's argument about obfuscation. There is no attempt to obfuscate or hide any aspect of the algorithm. If we wanted to obfuscate it, we would have just omitted Appendix C. Instead, Appendix C was referenced twice in the main paper, even before the revision. This is the proof of our good faith. Moreover, in the revision, we added a comment in the "Sample Collection" paragraph to state that the performance improvement guarantees are ensured for deterministic environments only. In addition, in the abstract, intro, and setup, we never claim that we provide an RL approach endowed with general performance improvement guarantees; rather, we highlight the interest in studying the connection between variance reduction and performance improvement. We will further stress the limitations of the performance improvement guarantees in these sections.
> > >
> > > >**While the paper is introduced and framed as a strategy for RL, most of the presentation focuses on an "unconstrained" setting where the distribution over trajectories is not constrained by the MDP.**
> > >
> > > We disagree with this statement. Sections 3 and 4 are devoted to the unconstrained setting, for a total of little more than 2 pages. Section 5 is devoted to the constrained setting that amounts little less than 2 pages. Moreover, the algorithm in Section 6 is presented in the constrained setting. Therefore, we devoted a similar space for the two settings.
> > >
> > >
> > > >**In addition to be a very confusing way to present the algorithm, this obscures some fundamental issues with the approach that are not at all discussed outside of an appendix.**
> > >
> > > We disagree that our choice represents “a very confusing way to present the algorithm” since, as we already highlighted in the first answer, the generality of the presentation is a point of strength, in our view, since it shows that the approach can be employed even outside RL.
> > >
> > > It is not true that these “fundamental issues … are not at all discussed outside of an appendix”. Instead, we discussed these issues twice in the main paper. First, for the general constrained setting, in the paragraph below Theorem 5.2, we have specified that the performance improvement guarantees are limited to certain conditions. At the end of this paragraph, we also reference Appendix C. These considerations were present even before the revision. Second, for the RL setting, we added in the revision a sentence in the “Sample Collection” paragraph in which we explicitly state that the performance improvement guarantee is ensured for the deterministic case only and we reference Appendix C. Due to space constraints, we couldn’t fit Appendix C in the main paper.
> > >
> > > >**Specifically, the improvement guarantee seems to only hold in deterministic environments unless we have a good estimate of the expected return conditioned on a trajectory of actions.**
> > >
> > > As we pointed out in the first answer, conditioning on the trajectory actions is a theoretical way to extend the performance improvement guarantees to the stochastic setting, not a way of implementing the algorithm. As we pointed out when submitting the revision, we tested the parameter-based version of our algorithm with increasing stochasticity of the environment and the results are shown in Appendix E.1. We can clearly see that our approach is able to outperform POIS in the case of large stochasticity (Figure 5). We conjecture that this good performance is due to the implicit trust region that compensates for the possible bias introduced by the missing performance improvement guarantee.

---

> > > > ### Author Response · Authors · 2021-11-27
> > > > **Re: post rebuttal comment (II)**
> > > >
> > > > >**Second, I am not totally convinced by the empirical evaluation for a few reasons. First, in the main experiment (Figure 2), the proposed algorithm seems to be outperformed by the baseline PB-POIS.**
> > > >
> > > > This is true for the Cartpole environment only, whereas for the other 3 environments, PB-POIS has performance almost overlapping with either AB-PO2PE or PB-PO2PE.
> > > >
> > > > >**Second, it seems the authors tune twice as many hyperparameters for their algorithm compared to the baselines.**
> > > >
> > > > Our algorithm natively comes with 4 hyperparameters ($J$, $\delta$, the step size, the variance initialization) while for TRPO there are no hyperparameters with the same meaning of $J$ (since there is no inner loop in TRPO) and of $\delta$ (since TRPO does not employ any confidence bound). We consider as appropriate an approach that tunes the hyperparameters available for each of the considered algorithms.
> > > >
> > > > >**Third, all the experiments are only done on small environments with linear models. I recognize that the main contribution of the paper is perhaps not to scale up the algorithm, but this is still a clear weakness of the evaluation to me.**
> > > >
> > > > As we already pointed out in the first answer, devising a practical algorithm is out of the scope of this paper. This paper tackles a novel view of importance sampling, focusing on its theoretical properties and providing a first algorithm proposal.
> > > >
> > > > >**I see that perhaps if we restrict to action-based algorithms the improvement of POPE over the baseline is more clear, but am not sure how this restriction would be justified. The robustness to batch size is potentially interesting if it indeed shows lower variance of the gradient, but I think it is still a bit unclear as presented because the smaller batch sizes are not actually leading to dramatically improved sample complexity in the main experiments (perhaps because there is no comparison to PB-POIS in the batch size experiments).**
> > > >
> > > > The reviewer's issue about the significance of this experiment is not clear to us. If the issue is about the fact that results are not "dramatically improved", we disagree. In Figure 3, it is clear that moving from $n=50$ to $n=11$, POIS significantly degrades in performance (blue curve) while PO2PE is still able to achieve the optimal performance (red curve). If the issue is about the fact that the parameter-based version was not tested here, we just have had not enough time to run those experiments for the time available for the revision and, above all, we believe they are not necessary as the action-based ones already illustrate the robustness to small batch sizes.

---

> ### Author Response · Authors · 2021-11-20
> **Reply to Reviewer pKkx (I)**
>
> We thank the reviewer for the constructive feedback. We are glad that the reviewer has recognized as “novel and interesting” the idea of using importance sampling with potential “different strengths” compared with previous policy optimization methods.
>
> ***Weaknesses***
>
> 1. We decided to keep the level of presentation as general as possible to highlight that the approach can, in principle, be employed for the stochastic optimization of general distributions even outside RL. Nevertheless, we are aware that this could make the paper harder to understand. We are working on a rebuttal revision of the paper where we anticipate the RL context and provide more connections earlier in the paper. We hope, with this, to facilitate the understanding of the paper.
>
> 2. We thank the reviewer for giving us the opportunity to clarify this point. In all our experiments, the function $f$ is taken to be the trajectory return $\mathcal{R}(\tau)$, regardless of the stochasticity of the environment. We were able to provide the performance improvement guarantee only when the underlying environment is deterministic. Indeed, in such a case, we are in the unconstrained setting as the trajectory distribution takes the form:
>
> $$
> p(\tau|\boldsymbol{\theta}) = \prod_{t=0}^{H-1} \pi_{\boldsymbol{\theta}}(a_i|s_i) \qquad \text{where } s_{t+1} = P(s_t,a_t),
> $$
>
> and as a result, thanks to Proposition 4.1, we have a performance improvement. Instead, in the stochastic setting, we have all transition models factors making the setting unavoidably constrained:
>
> $$
> p(\tau|\boldsymbol{\theta}) = \prod_{t=0}^{H-1} \pi_{\boldsymbol{\theta}}(a_i|s_i) P(s_{t+1}|s_t,a_t)
> $$
>
> Consequently, due to Theorem 5.2 the algorithm no longer has the performance improvement guarantee for every function $h$. Nonetheless, our experiments are performed with $\mathcal{R}(\tau)$ as function $f$.
>
> The content of Appendix C has to be intended as a theoretical way to make the performance guarantee of Proposition 4.1 hold even in stochastic environments, reducing the problem to the unconstrained case, rather than a way to implement the algorithm. Indeed, estimating $\mathbb{E}[\mathcal{R}(\tau)|\underline{a}]$ is quite challenging. We admit that guaranteeing performance improvement for deterministic environments only is a limitation of the approach, but we think it is a price to pay for achieving the implicit trust region. Indeed, we conjecture that although the objective function introduces a bias (the lack of performance improvement guarantee), this is compensated by the significant variance reduction that is a consequence of the implicit trust region.
>
> We are also complementing the experimental evaluation with simulations in the parameter-based setting [1], in which stochasticity is shifted into the hyperpolicy from which we sample the parameters of a deterministic policy. This way, the involved distribution would be:
>
> $$
> \nu_{\boldsymbol{\rho}}(\boldsymbol{\theta}) p(\tau|\boldsymbol{\theta}),
> $$
>
> where $\boldsymbol{\rho}$ are the parameters of the hyperpolicy. In this way, drawing a parallel with Appendix C, even in the presence of stochastic environments, to obtain the unconstrained setting ensuring the performance improvement, we just need to have an estimate of $J(\boldsymbol{\theta}) = \mathbb{E}[\mathcal{R}(\tau) | \boldsymbol{\theta}]$, which can be obtained with a simple critic and it is indeed easier to estimate than $\mathbb{E}[\mathcal{R}(\tau)|\underline{a}]$. We are adding these considerations to the rebuttal revision.
>
> [1] Sehnke, Frank, Christian Osendorfer, Thomas Rückstieß, Alex Graves, Jan Peters, and Jürgen Schmidhuber. "Policy gradients with parameter-based exploration for control." In International Conference on Artificial Neural Networks, pp. 387-396. Springer, Berlin, Heidelberg, 2008.

---

### Official Review · Reviewer_P7sd · 2021-11-08

**Correctness:** 4
**Technical Novelty And Significance:** 3
**Empirical Novelty And Significance:** 2
**Recommendation:** 6
**Confidence:** 4

**Main Review:**

## Strong points:
Significance and Novelty: The paper is motivated by the technical tools of importance sampling (IS). The starting point of the paper is the difference of using IS in different communities, in RL community, IS is served as a **passive** tool but in Monte Carlo simulation community the behavior policy of IS can be **actively** picked and learned to reduce the variance. The minimum-variance behavior distribution can also be served as an improved policy compared with the original one. The paper later discuss how this minimum-variance policy can connect with the trust region improvement by discussing several theorems with the Renyi's Divergence.
I enjoy reading those connections and I think the discussion of the minimum-variance policy reveals a new way to thinking trust region/safe RL algorithm, which can draw a potential interest of the RL community.

Technical Clarity: The paper is well-organized the easy to follow. The mathematical formula is clear and strict.

Soundness: all the proof is sound to me.

## Weak points:
Empirical Clarity: I feel like the final algorithm is not that clear from the first pass of my reading especially the inner loops of the algorithm (see detailed questions below).

Empirical Evaluation: The design of the experiment is fine and clear to me, but I would expect for more ablation study coming up along the the detailed discussion in the previous Section, such as the choices of $\alpha$ and more choices of h (such as Ackley function you mentioned in Figure 1).  I would feel like the empirical results does not fully support the theoretical discussion.

The Original Motivation: Most of the paper (section 4-5) discussed the policy improvement property of the minimum-variance policy. But I think the **variance** itself is as importance as a topic you might also want to highlight, as you mentioned the robustness to small batch size in the experiment. I would expect a paragraph to make this point more clear (for now from Eq. (6) I could have a vague sense why the algorithm is robust to small batch size)

## Questions:
Here are a few questions I would like the authors to answer to clarify my understanding.

1. Does Eq. (6) your final objective? Or do you include the finite sample upper bound in Theorem 6.1 as well? If you use Eq, (6), how do you guarantee a minimization of Line 5 in your algorithm? It seems to me in Appendix D you mention a closed form for Gaussian, is that the closed form for minimization of Eq. (6) (or Theorem 6.1)? Or do you just use several steps of stochastic gradient descent?

2. It seems to me that in the experimental session you fix $\alpha = 2$. Have you tried different $\alpha$? How will $\alpha$ affect the robustness of batch sizes.

3. The model assumption you use for TRPO, POIS and you algorithm is a linear Gaussian policy. What happened if you use a more complex policy class such as neural network? Will your algorithm still outperform TRPO in this case?

4. Most of the analysis in your algorithm is not specific to RL. But in RL, there is a famous "curse of horizon" phenomenon in long horizon setting [1] when your importance ratio is using the product of policy ratio for each time step. Is that a problem in your experiment? How long is the horizon you use in your experiments?

## Minors:
In the paragraph under Theorem 4.2, "all three properties" should be "both properties" (I did not see 3 properties in Theorem 4.2).

References:
1. Liu, Q., Li, L., Tang, Z., & Zhou, D. (2018). Breaking the curse of horizon: Infinite-horizon off-policy estimation. arXiv preprint arXiv:1810.12429.

**Summary Of The Paper:**

The paper proposes a policy improvement algorithm inspired by the minimum-variance policy of importance sampling (IS) technique in Monte Carlo simulation community. Properties of the new policy improvement algorithm such as convergence and implicit trust region are well studied in the paper. And in practical, the paper leverages a surrogate objective with the non-central alpha-moment as the finite sample objective for the algorithm. The new algorithm is used in the policy optimization in reinforcement learning. Empirical experiments on several continuous control demonstrate the benefit of the new proposed algorithm compared with existing trust region baselines, especially on the robustness to the small batch sizes.

**Summary Of The Review:**

Although there are several points need to be addressed by the authors, I feel the theoretical contribution is strong enough to recommend at least a borderline acceptance at this point. I am open to raise my score if my questions can be addressed.

---

> ### Author Response · Authors · 2021-11-20
> **Reply to Reviewer P7sd**
>
> We thank the reviewer for appreciating our new way of thinking about trust-region/safe RL algorithms and for recognizing the potential interest in the RL community. We address the issues in the following.
>
> ***Weak Points***
>
> [Empirical Clarity] We are aware that the final algorithm might not be completely clear. We are currently working on a rebuttal revision to improve this part.
>
> [Empirical Evaluation] We performed a study on the value of $\beta$, which corresponds to the exponent of different choices of $h$ as power function (e.g., $h(x) = x^\beta$), like the one performed for the Ackley function (Figure 1). This experiment has been conducted on the Inverted Double Pendulum environment and the results are shown in Figure 4.
>
> [The Original Motivation] We agree with the reviewer. The variance reduction is a very relevant aspect of our approach. The variance reduction property is somehow implicit, as our approach is initially designed to find the minimum-variance behavioral policy and only afterwards we study the properties of this minimum-variance behavioral policy in terms of performance improvement. We will stress this point in the revision.
>
> ***Questions***
>
> 1. Our final objective is the one presented in Theorem 6.1, which includes the estimator and the finite-sample penalization. The closed-form for Gaussian distributions refers to the integral inside the square root only, while the complete objective is optimized via several steps gradient descent. We are updating the algorithm for the rebuttal revision specifying the actual objective implemented.
>
> 2. We tested just the value $\alpha=2$, but we considered different transformation functions $h$ in Figure 2. See also Answer to [Empirical Evaluation].
>
> 3. As mentioned in the *General Comment*, the goal of this paper is to shed light on a novel view of importance sampling, showing preliminary empirical evaluations able to motivate the study of this direction. We did not intend to provide a practical algorithm able to display competitive performance with more complex policy classes. We think this should be considered a future extension and out of the scope of this paper.
>
> 4. We employed a horizon equal to 500 (we will add this in the Appendix). We agree that in policy optimization, the curse of the horizon is an issue, but we think that this strengthens the need for employing a variance reduction approach like the one we propose in this work. Empirically, we have not detected problematic behaviors and we ascribe this to the beneficial effect of the implicit trust region.
>
> ***Minors***
> We have fixed the typo.

---

### Author Response · Authors · 2021-11-20
**Rebuttal**

We would like to thank the reviewers for their insightful comments and suggestions. We are happy that all reviewers appreciated the novelty and the significance of our new active view of off-policy learning. We hope that the rebuttal and the revised paper effectively address the raised concerns.

**General Comment**

We are aware that the experimental evaluation is conducted in low-dimensional environments and with simple policy models. Nevertheless, we stress that the main purpose of our work is to shed light on an appealing view of importance sampling, i.e., the one related to variance reduction, which is typically disregarded in the reinforcement learning literature, focusing on its theoretical relations with policy improvement. For these reasons, we privileged the simplicity of experiments, believing that the experimentations on high-dimensional environments and more complex policies (e.g., neural networks) are more suitable for future practical versions of the algorithm.

We are currently working on a paper rebuttal revision that incorporates the reviewers’ suggestions about the clarity of the algorithm presentation and that provides additional experimental results. We will submit it as soon as possible.

---

> ### Author Response · Authors · 2021-11-22
> **Revision Sumbitted**
>
> Dear reviewers,
>
> we have submitted the rebuttal revision of our paper. The modifications are highlighted in purple in the paper and they can be summarized as follows:
>
> - We changed the description of the algorithm (Section 6) to clarify its structure better, and we have modified the pseudocode (Algorithm 1) in order to include the actual objective that is optimized (the one in Theorem 6.1).
> - We stressed the benefits of variance reduction in terms of robustness to the batch size in the Discussion and Conclusions (Section 8).
> - We included in the paper the parameter-based version (PB) of the algorithm (in addition to the action-based (AB) one already present). This modification mainly affects Sections 6, in which the "Sample Collection" paragraph has been extended, and Section 7. For Section 7, we added the experimental results comparing the parameter-based version of our approach (PB-PO2PE) and the parameter-based version of POIS (PB-POIS). In Appendix E, we added the corresponding hyperparameters. We extended Appendix C to include this case too.
> - We added in Appendix E.1 an experiment to show that our approach (in the parameter-based version) is competitive even in the case of stochastic environments, in which the performance improvement guarantee no longer holds.
> - We added in Appendix E.2 some considerations about the translation operation needed in case of negative reward.
> - We have fixed the typos.
>
> We hope that the rebuttal, together with the revised paper, will clarify the raised issues.

---

### Decision · Program_Chairs · 2022-01-20

**Decision:**

Reject

**Comment:**

This paper received a split of scores, from 3 to 10. Among the reviewers, there are both strong advocates and strong rejects. All reviewers agree that finding a policy that is not only improving value but also has lowered variance is an interesting ideas. However, many reviewers point out that are major clarity issues that might hide fundamental problems. The proved guarantees seem to require strong assumptions that are unlikely to hold in practice, and experimental comparisons also have some subtleties. Taking together, although this could be a very interesting work, it will require a major revision and another round of review+discussions before it can be shaped into an acceptable paper.